# Subtle Errors Matter: Preference Learning via Error-injected Self-editing

## Abstract

Large Language Models (LLMs) have exhibited strong mathematical reasoning and computational prowess, tackling tasks ranging from basic arithmetic to advanced competition-level problems. However, frequently occurring subtle errors, such as miscalculations or incorrect substitutions, limit the models' full mathematical potential. Existing studies to improve mathematical ability typically involve distilling reasoning skills from stronger LLMs or applying preference learning to step-wise response pairs. Although these methods leverage samples of varying granularity to mitigate reasoning errors, they overlook the frequently occurring subtle errors. A major reason is that sampled preference pairs involve differences unrelated to the errors, which may distract the model from focusing on subtle errors. In this work, we propose a novel preference learning framework called eRror-Injected Self-Editing (RISE), which injects predefined subtle errors into partial tokens of correct solutions to construct hard pairs for error mitigation. In detail, RISE uses the model itself to edit a small number of tokens in the solution, injecting designed subtle errors. Then, pairs composed of self-edited solutions and their corresponding correct ones, along with pairs of correct and incorrect solutions obtained through sampling, are used together for subtle error-aware DPO training. Compared with other preference learning methods, RISE further refines the training objective to focus on predefined errors and their tokens, without requiring fine-grained sampling or preference annotation. Extensive experiments validate the effectiveness of RISE, with preference learning on Qwen2-7B-Instruct yielding notable improvements of 3.0% on GSM8K and 7.9% on MATH.

## 1 Introduction

Large Language Models (LLMs) have demonstrated advanced logical reasoning and computational capabilities, enabling them to tackle a diverse range of problems from elementary mathematics to complex, competition-level questions (Yue et al., 2024a; Gou et al., 2024; Yu et al., 2024). Nevertheless, frequently occurring subtle errors remain one of the primary obstacles preventing these models from consistently realizing their full mathematical potential (Li et al., 2024). These errors, such as miscalculations, incorrect substitutions, and omission of calculation terms, account for approximately 75% of the total errors as shown in Figure 3.

Many existing works have attempted to improve mathematical ability by distilling reasoning skills from stronger LLMs using synthetic datasets (Shao et al., 2024; Yue et al., 2024a; Azerbayev et al., 2024). While distilling helps the model acquire expanded reasoning patterns, it does not directly enable it to reduce the occurrence of errors (Hong et al., 2024). Recently, a growing body of research has shifted its focus toward preference optimization with fine-grained preference pairs to mitigate mathematical errors (Lai et al., 2024; Chen et al., 2024; Setlur et al., 2024a). This method leverages Direct Preference Optimization (DPO) to reduce the likelihood of generating the error step in a pair of steps with the same context (Rafailov et al., 2023). However, learning with these fine-grained preference pairs still struggles to focus on subtle errors, as the aforementioned pairs involve some differences unrelated to the errors. Additionally, establishing the preference relationship within fine-grained pairs requires extra LLM-based annotations or approximate estimations, which may introduce bias (Xie et al., 2024; Lai et al., 2024).

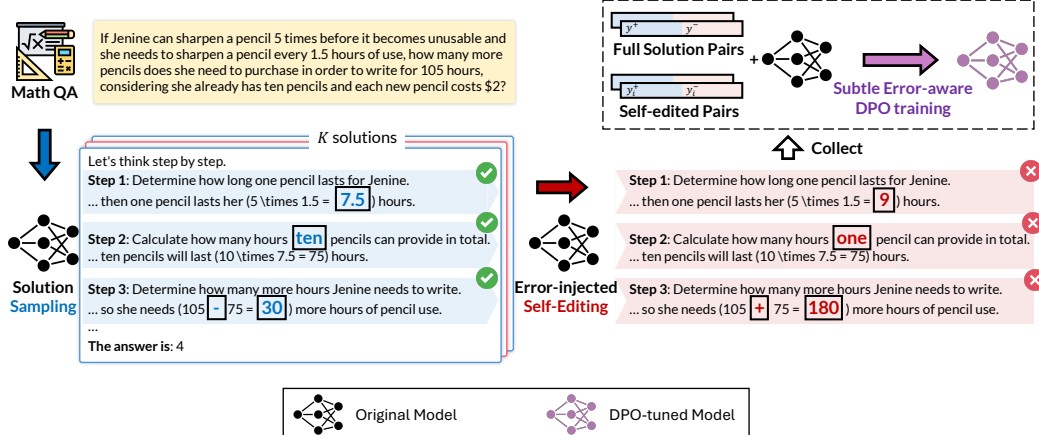

Figure 1: Preference learning framework augmented by error-injected self-editing. Each mathematical problem is sent to the original model to sample $K$ solutions, with correct and incorrect solutions in rectangles with blue and red borders. For one correct solution, we inject errors into each step of the solution and collect self-edited pairs. We also select an incorrect solution paired with the above correct one as full-solution pairs. Both sampling and self-editing are performed by the same model.

In our work, we argue that, in addition to sampling preference pairs and annotating their relationships, generating a correct solution and injecting errors into it can be an effective approach to construct pairs that specifically target subtle errors. First, error injection can introduce errors into a small number of tokens while preserving the overall structure of the solution. Such error-injected samples, with limited differences from the correct solutions, can be regarded as hard negatives for preference learning (Liu et al., 2024). Second, we can leverage the LLM itself to inject predefined errors using appropriate prompts. Compared to randomly sampled pairs, pairs based on injected errors are more controllable, allowing preference learning to focus on designed, targeted subtle errors.

Based on the above motivation, we propose a novel preference learning framework called eRror-Injected Self-Editing (RISE). The key idea is to prompt the LLM to inject errors into correct solutions and construct hard pairs targeting predefined subtle errors for preference learning. To be specific, we first apply an LLM to generate several multi-step solutions and construct a full-solution pair composed of one correct solution and one incorrect solution. Then, we choose the correct one and edit each step of the solution to inject subtle errors. The same model is used for error-injected editing, as it may better recognize the tokens where it is most prone to making mistakes. We design several types of subtle errors and modify a few tokens to introduce these errors. The edited steps and the corresponding correct steps are constructed as self-edited pairs. Finally, we conduct subtle error-aware DPO training on both self-edited pairs and full-solution pairs. Inspired by Pal et al. (2024), a negative log-likelihood loss is introduced to stabilize the training, as the self-edited pairs are highly similar, which can easily reduce the likelihood of the correct solutions.

We evaluate our framework on two LLM series: Qwen2 and Llama-3.1. Our method RISE-QWEN2-7B achieves a 3.0% accuracy gain on GSM8K and 7.9% on MATH, and RISE-LLAMA-3.1-8B achieves 3.9% and 2.7%, respectively. Detailed error analysis shows that RISE helps the LLM further avoid predefined subtle errors. Moreover, our method successfully generalizes reasoning preferences derived from mathematical tasks to other reasoning domains, such as logical reasoning and code generation.

## 2 METHOD

In this section, we introduce our RISE framework, which achieves hard negative preference pair construction through error-injected self-editing and enhances the mathematical reasoning capability of LLMs by subtle error-aware DPO training. As shown in Figure 1, our method starts with sampling $K$ multi-step solutions from the original model. One correct solution and one incorrect solution are chosen as a full-solution pair based on the correctness of the final answer. Next, we use the same

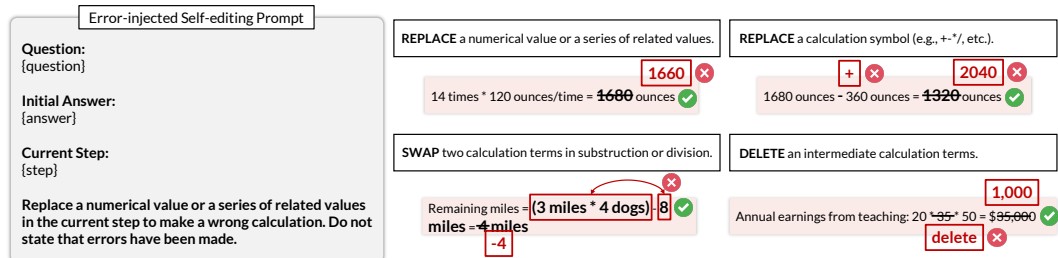

Figure 2: Error-injected self-editing prompt and some error injection examples. We display three error-injected self-editing operations: "REPLACE", "SWAP", and "DELETE".

model to edit each step of the correct one, injecting subtle errors into a small number of tokens (Sec. 2.1). The edited steps and the correct steps are collected as self-edited pairs. Finally, the combination of the self-edited pairs and full-solution pairs is employed for subtle error-aware DPO training (Sec. 2.2).

We adopt an instruction-tuned model in our experiments, since it can be used to sample a solution and self-edit it with appropriate prompts. Given a mathematical problem $x \in \mathcal{D}_{\text{raw}}$, we first prompt the model $M$ to sample a multi-step solution set $\{\hat{y}\}$ following Lai et al. (2024). To ensure that the sampled solution strictly follows an explicit Chain-of-Thought (CoT) structure, we prepend the model's generated solution with the prefix "*Let's think step by step. Step 1:*". This prefix ensures that each step of the solution begins with a "Step" marker. We select the solution with the output answer that matches the reference answer as the correct one for subsequent editing. The correct and the other incorrect are used to construct a full-solution pair set $\mathcal{D}_{\text{Full}}^{\pm}$. The correct solution can be denoted as $\hat{y}^+ = \bigoplus_{i=1}^{n} \hat{y}_i^+$, where $\hat{y}_i^+$ is the $i$-th step and $n$ represents the total number of steps.

## 2.1 DATASET CONSTRUCTION VIA ERROR-INJECTED SELF-EDITING

To create hard preference pairs, we employ an editing approach to modify certain tokens in the correct solution to introduce predefined errors. We focus on the error injection of each step in one solution. Compared with previous step-wise preference learning (Lai et al., 2024), our approach further refines the objective of preference learning by specifically targeting a few error tokens, enhancing the ability of LLMs to avoid subtle errors.

**Error Types.** As we aim to optimize the model to avoid subtle errors, the main types of errors we concentrate on are as follows: (1) Numerical calculation errors; (2) Numerical or symbolic substitution errors; (3) Omission of calculation terms. These errors frequently occur in most solutions and typically involve only a small amount of tokens. We also include two relatively less frequent errors: (4) Errors in the calculation order and (5) Errors in the use of calculation symbols. We extract and summarize these errors from the solutions generated by the models in our experiments.

**Self-editing.** We design appropriate prompts and utilize the model itself to edit the generated correct step $\hat{y}_i^+$. As most of the steps in the correct solutions are accurate (Lu et al., 2024b), we use all the steps from the correct ones without applying any additional filtering. Since solutions to mathematical problems are highly error-sensitive, any modification of numerical values, symbols, or other mathematical elements disrupts the original correct procedure. Thus, even small language models can be prompted to almost certainly inject errors. The error-injected step $\hat{y}_i^-$ is generated using an edit prompt shown on the left of Figure 2. This edit prompt contains the problem $x$, the concatenation of previous steps $\hat{y}_{<i}$, the correct step $\hat{y}_i$, and the edit type $e$. We mainly use three types of editing operations: "REPLACE", "SWAP", and "DELETE" (Mallinson et al., 2022). Some error injection examples are shown on the right of Figure 2. For each step, only around ten tokens will be modified. We collect the error-injected steps and the correct steps to construct a paired edited set $\mathcal{D}_0^{\pm} = \{(\hat{y}_i^+, \hat{y}_i^-)\}$. These edited pairs are then filtered through Levenshtein distance-based

---

**Algorithm 1 Preference Learning via Error-injected Self-editing**

---

**Input:** $\mathcal{D}_{\text{raw}}$: mathematical problems; $M$: original model; $\mathcal{E}$: edit prompt set; $K$: number of sampling attempts; $N$: number of self-edited pairs;

Initialize the subtle error-aware DPO training dataset $\mathcal{D}_M^{\pm} \leftarrow \{\}$

**for** $x \in \mathcal{D}_{\text{raw}}$ **do**

    Sample $K$ solutions $\{\hat{y}\} \sim P_M(\cdot \mid x)$.

    Randomly select one correct solution $\hat{y}^+$ and one incorrect solution $\hat{y}^-$.

    Define $\hat{y}^+ = \hat{y}_1^+ \oplus \hat{y}_2^+ \oplus \cdots \oplus \hat{y}_n^+$, where $n$ denotes the number of steps in the solution.

    Initialize self-edited pair set $\mathcal{D}_{\text{Edit}}^{\pm} \leftarrow \{\}$.

    **for** $i = 1$ **to** $n$ **do**

        Randomly select an edit type $e \leftarrow \mathcal{E}$.

        Edit the step using the same model $\hat{y}_i^- \sim P_M(\cdot \mid e, x, \hat{y}_{<i}^+, \hat{y}_i^+)$, where $\hat{y}_{<i}^+$ represents the concatenation of steps before $\hat{y}_i^+$.

        $\mathcal{D}_{\text{Edit}}^{\pm} \leftarrow \mathcal{D}_{\text{Edit}}^{\pm} \cup \{(\hat{y}_i^+, \hat{y}_i^-)\}$ if $LD(\hat{y}_i^+, \hat{y}_i^-) < \alpha$, where $LD$ is the function to calculate edit distance.

    **end for**

    Randomly select $N$ self-edited pairs $\{(\hat{y}_j^+, \hat{y}_j^-)\}_{j=1}^N \leftarrow \mathcal{D}_{\text{Edit}}^{\pm}$

    $\mathcal{D}_M^{\pm} \leftarrow \mathcal{D}_M^{\pm} \cup \{(\hat{y}^+, \hat{y}^-), (\hat{y}_1^+, \hat{y}_1^-), \ldots (\hat{y}_N^+, \hat{y}_N^-)\}$

**end for**

Optimize the subtle error-aware DPO loss in Equation 3 on $\mathcal{D}_M^{\pm}$ with $P_M$ as the reference policy.

---

similarity as follows:

$$\mathcal{D}_{\text{Edit}}^{\pm} = \{(\hat{y}_i^+, \hat{y}_i^-) \mid LD(\hat{y}_i^+, \hat{y}_i^-) < \alpha, (\hat{y}_i^+, \hat{y}_i^-) \in \mathcal{D}_0^{\pm}\}, \tag{1}$$

where $LD$ represents the function to calculate Levenshtein distance and $\alpha$ denotes the filter threshold. The filtered pairs $\mathcal{D}_{\text{Edit}}^{\pm}$ are used for subsequent subtle error-aware DPO training.

## 2.2 SUBTLE ERROR-AWARE DPO TRAINING

Given all self-edited pairs and full-solution pairs, we apply DPO to optimize the model (Rafailov et al., 2023). Specifically, we first randomly choose $N$ pairs from all self-edited pairs for each problem and combine them with the full-solution pair, which will be $N + 1$ pairs in total as the training set. To effectively learn subtle errors in each step, we adopt the step-wise DPO loss (Lai et al., 2024) for the self-edited pairs as it can focus on fine-grained preference learning, and general DPO loss for the full-solution pairs. Two loss functions are defined as follows:

$$\mathcal{L}_{\text{DPO-Edit}} = -\mathbb{E}_{(\hat{y}_i^+, \hat{y}_i^-) \sim \mathcal{D}_{\text{Edit}}^{\pm}}[\log \sigma(\beta \log \frac{\pi_\theta(\hat{y}_i^+|x; \hat{y}_{<i}^+)}{\pi_M(\hat{y}_i^+|x; \hat{y}_{<i}^+)} - \beta \log \frac{\pi_\theta(\hat{y}_i^-|x; \hat{y}_{<i})}{\pi_M(\hat{y}_i^-|x; \hat{y}_{<i})})].$$

$$\mathcal{L}_{\text{DPO-Full}} = -\mathbb{E}_{(\hat{y}^+, \hat{y}^-) \sim \mathcal{D}_{\text{Full}}^{\pm}}[\log \sigma(\beta \log \frac{\pi_\theta(\hat{y}^+|x)}{\pi_M(\hat{y}^+|x)} - \beta \log \frac{\pi_\theta(\hat{y}^-|x)}{\pi_M(\hat{y}^-|x)})], \tag{2}$$

where $\pi_\theta$ is the policy model and $\pi_M$ is the reference model. To mitigate the risk of optimization failure caused by the high similarity between paired samples (Pal et al., 2024), we additionally introduce a negative log-likelihood loss for the correct samples. This loss can help prevent the collapse of the probability of generating correct samples and is defined as $\mathcal{L}_{NLL}$. We present our subtle error-aware DPO loss that contains the above two objectives as follows:

$$\mathcal{L} = \mathcal{L}_{\text{DPO-Edit}} + \mathcal{L}_{\text{DPO-Full}} + \lambda \mathbb{1}_{[r<0]}\mathcal{L}_{NLL},$$

$$r = \log \frac{\pi_\theta(\hat{y}_i^+|x; \hat{y}_{<i}^+)}{\pi_M(\hat{y}_i^+|x; \hat{y}_{<i}^+)} \text{ or } \log \frac{\pi_\theta(\hat{y}^+|x)}{\pi_M(\hat{y}^+|x)} \tag{3}$$

where $\lambda$ is the weight to control the balance of two objectives, $r$ signals when to apply the NLL loss. Inspired by Pal et al. (2024), we design indicator $r$ to represent whether the generation probability of the policy model is lower than that of the reference model. We outline the full algorithm of our Preference Learning via Error-injected Self-editing in Algorithm 1.

## 3 EXPERIMENTS

### 3.1 EXPERIMENTAL SETUP

**Evaluation Datasets.** We evaluate our framework on three in-domain datasets, GSM8k (Cobbe et al., 2021), MATH (Hendrycks et al., 2021) and AQuA (Ling et al., 2017), along with three out-of-domain datasets, SVAMP (Patel et al., 2021), AIME24 (MAA, 2024), and Odyssey-MATH (Net-mind.AI, 2024). These six datasets span a broad spectrum of mathematical problems, ranging from basic arithmetic to advanced competition-level problems. The problems in these datasets, including tabular, free-form, and multiple-choice formats, ensure a robust evaluation of the model's mathematical reasoning ability. The detail of all datasets is described in Table 6 in the Appendix A.

**Baselines.** Our framework is compared with several LLMs performing well in mathematical reasoning. Two closed-source state-of-the-art LLMs: OpenAI's GPT-4o (OpenAI, 2024) and Anthropic's Claude-3.5-Sonnet (Anthropic, 2024). Three open-sourced general instruction-tuned LLMs: Mistral-7B-Instruct-v0.3 (Jiang et al., 2023), Qwen2 series (Yang et al., 2024a), and Llama-3.1 series (Meta AI, 2024). Five mathematically enhanced LLMs: DeepSeekMath-RL (Shao et al., 2024), Llemma (Azerbayev et al., 2024), ToRA (Gou et al., 2024), MAmmoTH (Yue et al., 2024a), and MathGenieLM (Lu et al., 2024a). Four additional LLMs under step-wise DPO optimization are also included: Step-DPO series (Lai et al., 2024), SVPO (Chen et al., 2024), MCTS-DPO (Xie et al., 2024), and SCDPO (Lu et al., 2024b). We display results with CoT prompts for most baseline LLMs. ToRA and MAmmoTH are designed with built-in capabilities for tool usage, so we report the best results with tools.

**Training Details.** We adopt mathematical problems used by Lai et al. (2024) for preference learning. The training dataset contains around 9K problems with corresponding correct step-by-step solutions. We discard these solutions and use only the problems to construct our training set. The problems are mainly from MetaMath (Yu et al., 2024) and AQuA Ling et al. (2017). Details are presented in Table 7. We select two open-source LLM series, Qwen2 (Yang et al., 2024a) and Llama-3.1 (Meta AI, 2024) as our base LLMs. We apply the instruction-tuned version of these models to sample solutions and meanwhile edit each step of the solution. The number of sampling attempts is set to 5, and the number of self-edited pairs is set to 1 for the Qwen2 series and 3 for the Llama-3.1 series. Since some problems cannot yield correct and incorrect preference pairs within a limited number of sampling attempts, we ultimately use only about 4.5K problems to construct preference pair combinations for training. Implementation details are presented in the Appendix B.

### 3.2 MAIN RESULTS

We report the main results on different mathematical datasets shown in Table 1 and Table 2. The former is from commonly used mathematical datasets published earlier, consisting of three in-domain and one out-of-domain datasets. The latter is from the recent competition-level datasets and both datasets are out-of-domain. Overall, we can see that RISE outperforms the SOTA model at different scales. These results highlight the potential of our framework to help the general LLM to be a mathematical generalist. On several datasets, RISE-QWEN2-72B and RISE-LLAMA-3.1-70B even outperform some closed-source LLMs.

Table 1 presents the results on the GSM8K, MATH, AQuA, and SVAMP datasets. Our framework enables the LLM to achieve noticeable improvements in mathematical reasoning compared to the corresponding backbone. Especially on MATH and AQuA, RISE-QWEN2-7B obtains 7.9% and 2.7% accuracy gain, and RISE-LLAMA-3.1-8B obtains 3.2% and 5.5%. RISE-QWEN2-7B outperforms other popular mathematical LLMs on all four datasets. Additionally, RISE performs better than the SOTA step-wise preference learning frameworks. Compared to Step-DPO, which shares the same backbone LLM and requires GPT-4-based annotations, RISE obtains better results without annotations. In detail, it achieves 4.1% higher accuracy on MATH, 6.7% higher on AQuA, and 2.9% higher on SVAMP. We scale our experiments on 70B/72B models and also observe around 1.0% accuracy gain on some evaluation datasets such as MATH and AQuA.

Table 2 displays the results for two complex, competition-level mathematical problems, AIME24 and Odyssey-MATH. We observe that both ToRA and MAmmoTH, even with 70B parameters,

| Model | Size | GSM8K | MATH | AQuA | SVAMP[†] |
|---|---|---|---|---|---|
| Closed-source Models | | | | | |
| GPT-4o | - | 96.0 | 78.1 | 82.2 | 94.3 |
| Claude-3.5-Sonnet | - | 94.9 | 68.5 | 77.5 | 92.9 |
| Open-source Models | | | | | |
| Mistral-7B-Instruct-v0.3 | 7B | 57.5 | 15.1 | 20.4 | 69.7 |
| Qwen2-7B-Instruct | 7B | 85.4 | 52.2 | 66.5 | 89.3 |
| Llama-3.1-8B-Instruct | 8B | 84.0 | 48.3 | 55.9 | 85.7 |
| DeepSeekMath-RL | 7B | 87.7 | 52.7 | 59.0 | 88.4 |
| Llemma | 7B | 36.4 | 18.0 | - | - |
| MAmmoTH | 7B | 53.6 | 31.5 | 44.5 | 67.7 |
| ToRA | 7B | 68.8 | 40.1 | 23.6 | 68.2 |
| MathGenieLM | 7B | 80.5 | 45.1 | - | 83.3 |
| Qwen2-7B-Step-DPO | 7B | **88.5** | 55.8 | 63.0 | 88.7 |
| SVPO | 7B | 81.7 | 59.5 | - | - |
| MCTS-DPO | 7B | 81.8 | 34.7 | - | - |
| SCDPO | 7B | 80.1 | 47.7 | 48.4 | 83.2 |
| RISE-QWEN2-7B | 7B | 88.4 (+3.0) | **59.9** (+7.9) | **69.7** (+3.2) | **91.6** (+2.3) |
| RISE-LLAMA-3.1-8B | 8B | 87.9 (+3.9) | 51.0 (+2.7) | 61.4 (+5.5) | 87.5 (+1.8) |
| Qwen2-72B-Instruct | 72B | 93.1 | 68.8 | 78.3 | 93.1 |
| Llama-3.1-70B-Instruct | 70B | 94.9 | 65.0 | 77.1 | 93.0 |
| MAmmoTH | 70B | 76.9 | 41.8 | 65.0 | 82.4 |
| ToRA | 70B | 84.3 | 49.7 | 41.3 | 82.7 |
| MathGenieLM | 70B | 88.4 | 51.2 | - | 87.7 |
| Qwen2-72B-Step-DPO | 72B | 94.0 | 70.8 | 77.5 | 93.5 |
| RISE-QWEN2-72B | 72B | 94.0 (+0.9) | **69.8** (+1.0) | **79.1** (+0.8) | **93.8** (+0.7) |
| RISE-LLAMA-3.1-70B | 70B | **95.1** (+0.2) | 66.1 (+1.1) | 77.7 (+0.6) | 93.3 (+0.3) |

Table 1: Comparison of results on different commonly used mathematical datasets. † represents out-of-domain datasets.

fail to solve any of the problems in AIME24, highlighting the difficulty of these problems. Our framework activates the mathematical potential of Qwen2-72B-Instruct and delivers 3.7% accuracy gains on the Odyssey-MATH dataset. Since the problems in the AIME dataset are highly complex and the model's answering failure is not due to subtle errors, RISE is unable to further improve accuracy. The failure of RISE-LLAMA-3.1-70B on Odyssey-MATH may be due to the fact that Llama-3.1-70B-Instruct is already fine-tuned on diverse, complex mathematical datasets, with its accuracy increasing from 36.4% in Llama-3.0 to 60.4% in Llama-3.1 (Netmind.AI, 2024). Preference learning on our relatively simple datasets may harm its original reasoning performance. Overall, the results on both in-domain and out-of-domain datasets demonstrate that our framework can help general LLMs consistently improve their mathematical reasoning abilities by avoiding subtle errors.

## 3.3 ABLATION STUDY

We demonstrate the effectiveness of our framework through different training settings as detailed below: (1) **w/o self-edited pairs**, which removes the supplemented edited pairs and trains the model with full-solution pairs. (2) **w/o full-solution pairs**, which trains the model with the edited pairs only. (3) **w/o NLL loss**, which removes the loss used for stabilizing training. Table 3 shows the results of different settings.

From the table, we can observe that either self-edited pairs or full-solution pairs are effective for preference learning to improve mathematical reasoning. Both types of pairs achieve similar results

| Model | Size | AIME24[†] | Odyssey-MATH[†] |
|---|---|---|---|
| Closed-source Models | | | |
| GPT-4o | - | 3/30 | 52.9 |
| Claude-3.5-Sonnet | - | 4/30 | 48.0 |
| Open-source Models | | | |
| ToRA | 70B | 0/30 | 26.8 |
| MAmmoTH | 70B | 0/30 | 15.7 |
| Qwen2-72B-Instruct | 72B | 4/30 | 45.7 |
| Llama-3.1-70B-Instruct | 70B | **7/30** | **60.4** |
| Qwen2-72B-Step-DPO | 72B | 4/30 | 50.1 |
| RISE-QWEN2-72B | 72B | 4/30 (+0/30) | 49.4 (+3.7) |
| RISE-LLAMA-3.1-70B | 70B | 7/30 (+0/30) | 58.9 (−1.5) |

Table 2: Comparison of results on recent competition-level mathematical datasets. † represents out-of-domain datasets.

| Method | GSM8K | MATH | Method | GSM8K | MATH |
|---|---|---|---|---|---|
| Qwen2-7B-Instruct | 85.4 | 52.2 | Llama-3.1-8B-Instruct | 84.0 | 48.3 |
| RISE-QWEN2-7B | **88.4** | **59.9** | RISE-LLAMA-3.1-8B | **87.9** | **51.0** |
| - w/o self-edited pairs | 88.3 | 58.2 | - w/o self-edited pairs | 86.8 | 49.9 |
| - w/o full-solution pairs | 88.0 | 58.1 | - w/o full-solution pairs | 86.6 | 50.3 |
| - w/o NLL loss | 88.2 | 59.4 | - w/o NLL loss | 87.4 | 50.7 |

Table 3: Ablation study of training settings.

on the GSM8K and MATH datasets. Moreover, the combination of these two types of pairs can raise the accuracy to a new peak. Compared with standard DPO training (w/o self-edited pairs), our framework outperforms by 1.8% on the MATH dataset with Qwen2-7B-Instruct; and by 1.1% on GSM8K and 1.2% on MATH with Llama-3.1-8B-Instruct. Besides, the NLL loss helps improve accuracy by about 0.3%.

### 3.4 SUBTLE ERROR ANALYSIS

To analyze the effect of our framework on specific error mitigation, we counted the number of errors generated by different models on the MATH dataset. In detail, given the problem, the generated solution, and the reference answer, we prompt GPT-4o to detect the first error in any solution and output the error type in the final. To verify GPT-4o's accuracy in detecting errors, we manually selected 50 random samples and checked for consistency in the identified errors. 46 (92%) of the samples were accurately detected with their error types, which is acceptable for conducting the complete analysis. We display the number of different errors made by the Qwen2-7B series in Figure 3. We can observe that numerical calculation errors account for approximately 60% of the total errors and subtle errors we defined for 75%. Compared with the standard DPO, our framework additionally reduces the number of predefined errors. Especially for numeric or symbolic substitution errors and omission of calculation terms, RISE reduces the number of errors, whereas standard DPO does not achieve this. In addition, other errors, mainly misunderstanding of problems or concepts, increase due to preference learning, but RISE still performs slightly better than standard DPO.

### 3.5 IMPACT ON GENERAL REASONING CAPABILITIES

To thoroughly analyze changes in LLMs' reasoning capabilities, we evaluate RISE-tuned models on out-of-domain tasks such as logical reasoning and code generation. We select one logical reasoning

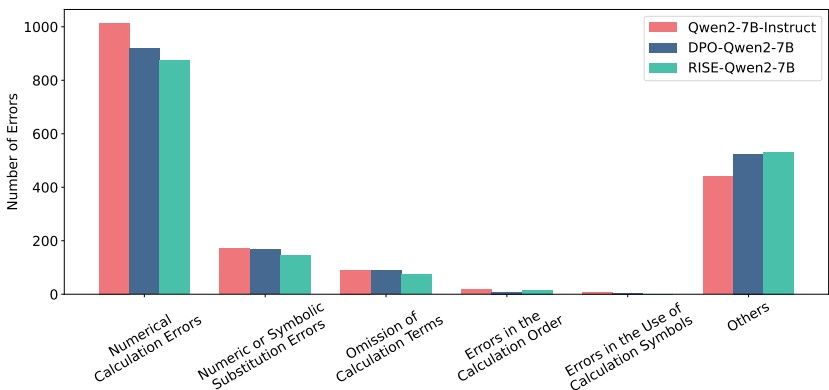

Figure 3: Error analysis across three models: Qwen2-7B-Instruct, DPO-Qwen2-7B, RISE-QWEN2-7B. We display the number of different types of errors when addressing the MATH dataset, where "Others" represents errors that fall outside the scope of consideration.

| Method | Puzzle | Cell | MBPP | Humaneval | Method | Puzzle | Cell | MBPP | Humaneval |
|--------|--------|------|------|-----------|--------|--------|------|------|-----------|
| | | Qwen2-7B | | | | | Llama-3.1-8B | | |
| Instruct | 8.1 | 21.5 | 42.2 | 43.9 | Instruct | 12.1 | 13.5 | 52.0 | 60.3 |
| DPO | 8.1 | 20.8 | 42.0 | 45.1 | DPO | 12.5 | 8.8 | 52.4 | 65.2 |
| RISE | **8.4** | **23.2** | **42.4** | **47.5** | RISE | **12.8** | **12.0** | **53.2** | **67.6** |

Table 4: Evaluation results on different out-of-domain tasks. "Puzzle" and "Cell" are abbreviations of Puzzle Accuracy and Cell Accuracy, which are metrics used in ZebraLogic benchmarks. "MBPP" and "Humaneval" are two code generation benchmarks, and we show pass@1 accuracy for each one.

benchmark, ZebraLogic, and two code generation benchmarks, MBPP and HumanEval for analysis. For ZebraLogic, we display Puzzle Accucary and Cell Accucary, and for MBPP and HumanEval, we display pass@1 accuracy. The models optimized with mathematical datasets (i.e., RISE-Qwen2-7B and RISE-Llama-3.1-8B) are used for evaluation. Table 4 presents the performance of mathematically RISE-tuned models on the two tasks mentioned above.

We can observe that, for the Qwen2-7B-Instruct and Llama-3.1-8B-Instruct models, RISE helps achieve accuracy increases in logical reasoning and code generation even without training on in-domain datasets. Moreover, RISE demonstrates superior performance compared to DPO, as DPO struggles to generalize reasoning capabilities to challenging out-of-domain tasks. Specifically, RISE-Qwen2-7B outperforms Qwen2-7B-Instruct in terms of Cell Accuracy and pass@1 accuracy on HumanEval, with improvements of 2.8% and 3.6%, respectively. RISE-Llama-3.1-8B achieves better pass@1 accuracy than Llama-3.1-8B-Instruct on MBPP and HumanEval, with improvements of 1.2% and 7.3%, respectively.

### 3.6 EFFECT OF SELF-EDITED PAIRS

Self-edited pairs are essential for fine-grained preference learning, particularly in mitigating subtle errors. To comprehensively explore the effect of self-edited pairs, we conduct experiments optimizing the model using different numbers of self-edited pairs (i.e., $N$ in Sec. 2.2). Figure 4 shows the results for $N = 1, 2, 3, 4$, and "All" self-edited pairs. For RISE-QWEN2-7B, the accuracies on GSM8K and MATH both decrease with more self-edited pairs. For RISE-LLAMA-3.1-8B, the accuracies reach a relative peak when using three self-edited pairs for each problem. This figure indicates that using more self-edited pairs is not always the better option, considering both the accuracy and the training cost of using additional samples. Additionally, RISE-LLAMA-3.1-8B prefers more self-edited pairs, which is consistent with the characteristics of Llama-3.1-8B-Instruct, as its full solutions contain around three more steps than those of Qwen2-7B-Instruct. More step-wise self-edited pairs help RISE-LLAMA-3.1-8B further avoid subtle errors.

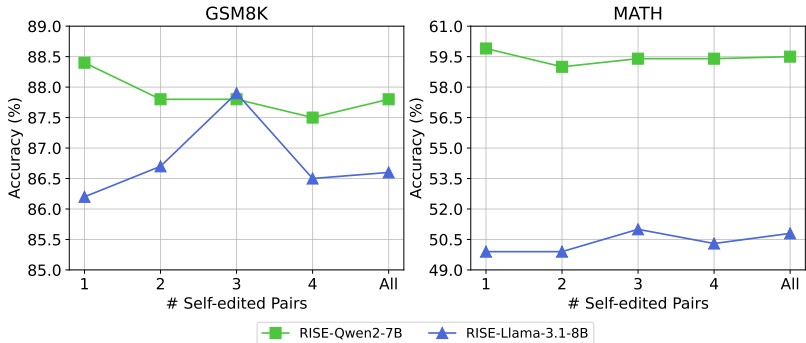

Figure 4: Effect of different numbers of self-edited pairs. The horizontal axis represents the number of self-edited pairs added for each problem during training, where "All" indicates the use of all self-edited pairs corresponding to a full solution.

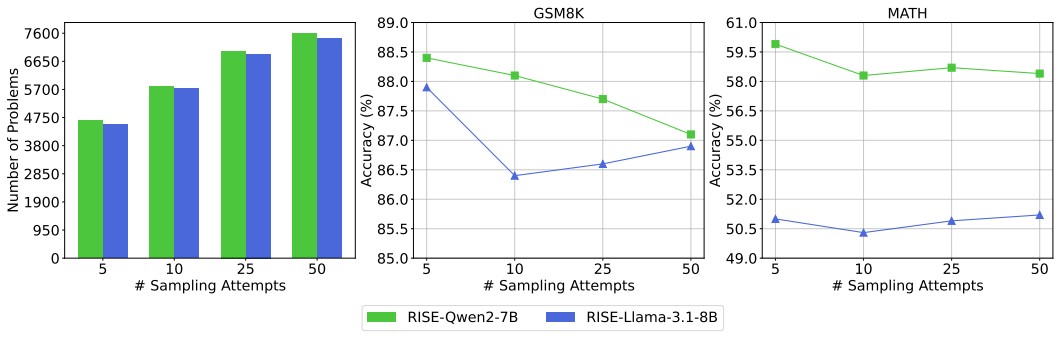

Figure 5: Effect of different numbers of sampling attempts. We sample multi-step solutions for a total of around 9K problems. The left figure shows the number of problems involved in training under different sampling attempts.

## 3.7 EFFECT OF SAMPLING ATTEMPTS

We further explore the effect of sampling attempts, which directly determine the number of problems involved in preference learning. As shown in Figure 5, as the number of sampling attempts increases, the number of problems corresponding to paired correct and incorrect full solutions also increases. Although more problems are involved in preference learning, the final results show that pairs corresponding to a larger number of problems actually reduce learning performance. It may be because more sampling attempts yield more samples of "extreme" problems. For these problems, the LLM tends to consistently answer either correctly or incorrectly. Training the LLM with these samples may not only be futile but could also lead to performance degradation. We observe this phenomenon on both RISE-QWEN2-7B and RISE-LLAMA-3.1-8B.

## 3.8 EFFECT OF DIFFERENT ERROR-INJECTION COMBINATIONS

Predefined injected errors are essential for the effectiveness of our framework. We investigate the impact of different combinations of injected errors on the model's mathematical performance. Three types of errors that occur most frequently in solutions generated by our method are selected for analysis. Since not all samples are applicable to these three types of errors, we focus on one primary error type (i.e., "**Dominate**"), supplemented by a small number of other errors. For example, some samples do not contain numerical values or calculation symbols, and thus cannot be injected with calculation errors. The comparison results are shown in Table 5. We can observe that all these combinations contribute to preference learning and a random combination yields the best performance.

| Combinations | GSM8K | MATH |
|---|---|---|
| Random | **88.4** | **59.9** |
| Calculation Errors Dominate | 88.2 | 59.0 |
| Substitution Errors Dominate | 88.1 | 58.6 |
| Omission of Calculation Terms Dominates | 87.8 | 58.7 |

Table 5: Effect of different error injection combinations for RISE-QWEN2-7B. "Calculation Errors" denotes numerical calculation errors, and "Substitution Errors" denote numeric or symbolic substitution errors.

It indicates that samples with diverse predefined errors are more likely to help the LLM learn to avoid subtle errors.

## 4 RELATED WORK

### 4.1 LLM FOR MATHEMATICAL REASONING

Large language models (LLMs) have shown remarkable proficiency in mathematical reasoning, excelling in tasks ranging from basic arithmetic questions to complex mathematical Olympiad problems (Team et al., 2023; Jiang et al., 2024; Dubey et al., 2024; OpenAI, 2024; Huang et al., 2024). Furthermore, various approaches have been investigated to enhance the mathematical reasoning capabilities of LLMs. Methods such as Llemma Azerbayev et al. (2024), DeepseekMath Shao et al. (2024), and Qwen2.5-Math Yang et al. (2024b) have focused on collecting vast amounts of math-related data for continued pretraining. Recent efforts have also focused on designing more efficient fine-tuning datasets to stimulate the mathematical capabilities of base models, such as MAmmoTH (Yue et al., 2024b), MetaMath (Yu et al., 2024) and DART-Math (Tong et al., 2024). Additionally, some works attempted to enhance the LLM's mathematical ability by incorporating external tools, particularly through the integration of code interpreter (Gou et al., 2024; Wang et al., 2024a; Liao et al., 2024).

### 4.2 STEP-WISE PREFERENCE LEARNING

In addition to pre-training and instruction fine-tuning, step-wise preference learning—particularly methods related to Proximal Policy Optimization (PPO)—has been widely explored to enhance the mathematical capabilities of LLMs (Lightman et al., 2024; Luo et al., 2023; Shao et al., 2024). However, the final performance is highly dependent on the quality of the process-supervised reward model (PRM) (Uesato et al., 2022; Wang et al., 2024b), and the training process of PPO or its variants is notably complex. Recently, Direct Preference Optimization (DPO) (Rafailov et al., 2023) simplified this process by directly leveraging pair-wise data for preference learning. Many works extend DPO, enabling it to perform step-wise preference learning to improve the model's multi-step mathematical problem-solving ability (Lu et al., 2024b; Xie et al., 2024; Setlur et al., 2024b; Lai et al., 2024). To the best of our knowledge, we are the first to tackle subtle errors in mathematical reasoning via step-wise preference learning especially on error tokens.

## 5 CONCLUSION

In this work, we propose a novel preference learning framework called eRror-Injected Self-Editing (RISE), which constructs hard pairs through self-editing to mitigate predefined subtle errors. Compared to other fine-grained, step-wise preference learning methods, our framework further refines the training objective to target subtle error tokens, without requiring LLM-based or estimation-based preference annotations. To avoid optimization failure caused by overly similar preference pairs, we additionally introduce a negative log-likelihood loss, controlled by the generation probability of the correct samples. The effectiveness of our framework is demonstrated in two LLM series: Qwen2 and Llama-3.1. Results across multiple mathematical datasets demonstrate that our method can unlock the model's potential in mathematical reasoning.

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

## A    EVALUATION SETTINGS

We apply all the baseline models and our RISE series model to generate solutions by greedy decoding (i.e., the temperature is set to 0). The vLLM framework with the 0.5.4 version is used to speed up decoding. For the GSM8K and MATH datasets, we use the evaluation script provided by the DeepSeek-Math project [1]. For other datasets, we use the evaluation agent provided by the odyssey-math [2].

Detailed evaluation dataset information is shown in Table 6.

| Eval Datasets | # Samples | In-Domain? | Answer Form |
|---|---|---|---|
| GSM8K (Cobbe et al., 2021) | 1319 | YES | Open-formed |
| MATH (Hendrycks et al., 2021) | 5000 | YES | Open-formed |
| AQuA (Ling et al., 2017) | 254 | YES | Multi-choice |
| SVAMP (Patel et al., 2021) | 1000 | NO | Open-formed |
| AIME24 (MAA, 2024) | 30 | NO | Open-formed |
| odyssey-math (Netmind.AI, 2024) | 387 | NO | Open-formed |

Table 6: Evaluation Datasets.

## B    IMPLEMENTATION DETAILS

We train 7B/8B models for 4 epochs with a global batch size of 96. The parameter $\beta$ is set to 0.4. For 70B/72B models, we train for 2 epochs with a global batch size of 64. The parameter $\beta$ is set to 0.5, and we use DeepSpeed ZeRO3 with CPU offload to reduce computational memory usage. The learning rate for all model training is set to 5e-7, and the parameter $\lambda$ is set to 0.05. We use Pytorch with the 2.4.0 version, Transformers with the 4.44.2 version, and deepspeed with the 0.14.4 version.

Detailed training dataset information is shown in Table 7.

| Eval Datasets | # Samples |
|---|---|
| GSM8K (Cobbe et al., 2021) | 1568 |
| MATH (Hendrycks et al., 2021) | 129 |
| MetaMath (Yu et al., 2024) | |
| - rewriting from GSM8K | 1387 |
| - rewriting from MATH | 953 |
| AQuA (Ling et al., 2017) | 4851 |
| Total | 8888 |

Table 7: Training Datasets.

## C    VALIDATION ON MORE OPEN-SOURCE MODELS

To further validate the effectiveness of the RISE framework, we implement additional experiments on Ministral-8B-Instruct and Qwen2.5-7B-Instruct, as these models are the most recent and well-regarded for their performance in various reasoning tasks. For Ministral-8B-Instruct, we sample 5 times and collect 7743 pairs of chosen and rejected samples, including a total of 3872 problems. For Qwen2.5-7B-Instruct, we sample 10 times and collect 5496 pairs of chosen and rejected samples, including a total of 2748 problems. The results are shown in the Table 8.

---

[1] https://github.com/deepseek-ai/DeepSeek-Math
[2] https://github.com/protagolabs/odyssey-math

| Method | GSM8K | MATH | Method | GSM8K | MATH |
|---|---|---|---|---|---|
| Ministral-8B-Instruct | 86.35 | 53.62 | Qwen2.5-7B-Instruct | 91.81 | 74.36 |
| DPO-Ministral-8B | 86.95 | 54.18 | DPO-Qwen2.5-7B | 92.49 | 75.00 |
| RISE-MINISTRAL-8B | **88.62** | **54.86** | RISE-QWEN2.5-7B | **92.95** | **75.06** |

Table 8: Results on Ministral-8B-Instruct and Qwen2.5-8B-Instruct.

# D    VALIDATION ON ANOTHER TRAINING DATASET

To evaluate our framework on a broader set of datasets, we have implemented additional experiments using other mathematical datasets, including problems from the original training sets of the GSM8K Cobbe et al. (2021) and MATH Hendrycks et al. (2021) datasets. We collect 15K problems like DART-math Tong et al. (2024) to conduct RISE training. The results on Qwen2-7B-Instruct indicate that our RISE framework achieves better performance than the general DPO method.

| Method | GSM8K | MATH |
|---|---|---|
| Qwen2-7B-Instruct | 85.4 | 52.2 |
| DPO-Qwen2-7B | 87.7 | 57.5 |
| RISE-QWEN2-7B | **88.6** | **58.5** |

Table 9: Results on another training dataset.

# E    EFFECT OF HYPERPARAMETER

We compare different values of the hyperparameter $\alpha$. The results of RISE-QWEN2-7B are shown in Table 10.

We can observe that an excessively large $\alpha$ may reduce the model's generalization ability, which in turn results in lower accuracy on GSM8K and MATH.

| $\alpha$ | 0.01 | 0.05 | 0.1 | 0.2 |
|---|---|---|---|---|
| GSM8K | 88.5 | 88.4 | 87.9 | 87.7 |
| MATH | 59.3 | 59.9 | 59.6 | 59.3 |

Table 10: Results of RISE-Qwen2-7B with different hyperparameter $\alpha$.

# F    EFFECT OF PROMPT DESIGN

To reduce reliance on manual prompt engineering and demonstrate the flexibility of prompts used in RISE, we use the self-instruct method to generate a variety of prompt templates (10 templates for each type of error) and conduct self-editing with a random choice of the generated prompts. Some examples of prompt templates are shown in Table 11 and Table 12.

With a random selection of prompt templates, our RISE can still help improve mathematical reasoning capability and outperform the general DPO method, as shown in Table 13. Compared with the results of the manual prompts used in our paper, the results of self-instruct prompts show a better accuracy on GSM8K but a slightly worse accuracy on MATH.

Besides, to further illustrate that our approach has the potential to be generalized to more diverse errors, we implement another experiment with a more universal prompt template. The prompt template is "Edit the current step to introduce an error. Do not state that errors have been made." This prompt doesn't indicate any error types and leverages the LLM itself to randomly introduce an error, which can capture broader spectrum error types. More importantly, this prompt can introduce arbitrary errors and even unexposed errors. The results on Qwen2-7B-Instruct with these self-edited samples are shown in Table 14.

**REPLACE a numerical value**

(1) Change a number in this step so that the calculation becomes incorrect, without indicating that a mistake has been introduced. (2) Alter the numerical value in this stage to produce an incorrect result, but avoid mentioning the error.

(3) Modify a number in the current calculation to lead to a wrong outcome, without revealing the inaccuracy.

(4) Adjust one of the values in this step to ensure the calculation is wrong, without pointing out the error.

(5) Replace a number in the calculation with an incorrect one, but do not mention that anything is wrong.

(6) Change a figure at this point to cause an erroneous result, without disclosing that you've made a mistake.

(7) Introduce a wrong number in this calculation step, but refrain from stating that an error has occurred.

(8) Modify a numerical value here so that the result is incorrect, without drawing attention to the mistake.

(9) Adjust the number in this step to generate an inaccurate result, without acknowledging the error.

(10) Introduce an incorrect value in this calculation, but avoid mentioning that the outcome is wrong.

Table 11: Prompts generated by the self-instruct method.

**SWAP two calculation terms**

(1) Switch the positions of two terms in the current calculation step to lead to an incorrect result, without explicitly acknowledging the mistake.

(2) Rearrange two terms in the present step in a way that causes an error, but avoid mentioning that a mistake has occurred.

(3) Alter the order of two terms in the current calculation to produce an incorrect outcome, without pointing out the error.

(4) Exchange the positions of two terms in this step to intentionally create a miscalculation, and don't indicate that anything is wrong.

(5) Adjust the placement of two terms in the ongoing calculation to introduce an error, without drawing attention to the fact.

(6) Swap the order of two terms in the current process to result in a wrong answer, but refrain from noting the mistake.

(7) Change the arrangement of two terms in the current step in a way that leads to an incorrect result, without signaling any error.

(8) Interchange two terms in the current calculation step to produce a mistake, while keeping the error implicit.

(9) Shift the positions of two terms in the calculation to create a wrong result, without stating that something is incorrect.

(10) Modify the sequence of two terms in this step, causing an incorrect calculation, but don't mention the flaw.

Table 12: Prompts generated by the self-instruct method.

# G  APPLICATION TO CODE GENERATION

To validate the effectiveness of our RISE framework on other reasoning tasks, we apply RISE to code generation. Following Xu et al. (2024), we adopt the LeetCode dataset[3] to conduct training. The dataset includes around 2K leetcode tasks in the medium and hard levels. For the Qwen2-7B-Instruct model, we sample 50 times and obtain 873 pairs of chosen and rejected full-solution samples for training. Then, we edit each chosen sample with the prompt "Edit the current step to introduce

---

[3]https://huggingface.co/datasets/greengerong/leetcode

| Method | GSM8K | MATH |
|---|---|---|
| RISE-prompt-manual | 88.4 | 59.9 |
| RISE-prompt-self-instruct | 88.6 | 59.3 |

Table 13: Results with the self-instruct prompts.

| Method | GSM8K | MATH |
|---|---|---|
| RISE-prompt-manual | 88.4 | 59.9 |
| RISE-prompt-arbitrary | 88.3 | 59.7 |

Table 14: Results with the arbitrary prompts.

an error. Do not state that errors have been made." This prompt can introduce arbitrary errors and can be easily adapted to other domains, such as code generation. Finally, we collect 1473 samples for RISE training. The results are shown in Table 15. We can observe that our RISE performs better than the general DPO method, achieving a 0.8% improvement on the MBPP test set and a 1.3% improvement on the Humaneval test set.

| Method | MBPP | Humaneval |
|---|---|---|
| Qwen2-7B-Instruct | 42.2 | 43.9 |
| DPO-Qwen2-7B | 43.4 | 46.3 |
| RISE-Qwen2-7B | **44.2** | **47.6** |

Table 15: Results on code generation.

# H CASE STUDY

## H.1 CASE STUDY ON QWEN2-7B SERIES MODELS

Table 16 and Table 17 shows a generated solution by RISE-QWEN2-7B and Qwen2-7B-Instruct. We can observe that the solution by Qwen2-7B-Instruct makes a mistake with calculation symbols, where the symbols in the red box should all be converted. RISE-QWEN2-7B successfully avoid this subtle error.

## H.2 CASE STUDY ON LLAMA-3.1-8B SERIES MODELS

Table 18 and Table 19 shows a generated solution by RISE-LLAMA-3.1-8B and Llama-3.1-8B-Instruct. We can observe that the solution by Llama-3.1-8B-Instruct makes a mistake with numeraical substitution, where the numerator should be $\sqrt{2}$ rather than $2\sqrt{2}$. RISE-LLAMA-3.1-8B successfully avoid this subtle error.

# I EDIT PROMPT SET

We show prompts for the aforementioned five types of subtle errors in Table 20, Table 21, Table 22, Table 23, and Table 24.

**Problem 1**:
What is the maximum value of $4(x + 7)(2 - x)$, over all real numbers $x$?
**Reference Answer**:
81

**RISE-QWEN2-7B**:
To find the maximum value of the function $f(x) = 4(x + 7)(2 - x)$, we can start by expanding the function and then completing the square or using calculus to find the critical points.
### Step 1: Expand the function

$$f(x) = 4(x + 7)(2 - x)$$
$$= 4(2x - x^2 + 14 - 7x)$$
$$= 4(-x^2 - 5x + 14)$$
$$= -4x^2 - 20x + 56$$

### Step 2: Find the critical points To find the maximum value, we can use calculus. We'll find the derivative of $f(x)$ and set it to zero to find critical points.

$$f'(x) = \frac{d}{dx}(-4x^2 - 20x + 56)$$
$$= -8x - 20$$

Setting the derivative equal to zero to find critical points:

$$-8x - 20 = 0$$
$$8x = -20$$
$$x = -\frac{20}{8} = -\frac{5}{2}$$

### Step 3: Determine the nature of the critical point To determine if this point is a maximum, we can use the second derivative test or simply recognize that since the coefficient of $x^2$ in the original function is negative, the parabola opens downwards, indicating that the critical point is indeed a maximum.
### Step 4: Find the maximum value Substitute $x = -\frac{5}{2}$ back into the original function to find the maximum value:
$$f\left(-\frac{5}{2}\right) = -4\left(-\frac{5}{2}\right)^2 - 20\left(-\frac{5}{2}\right) + 56$$
$$= -4\left(\frac{25}{4}\right) + 50 + 56$$
$$= -25 + 50 + 56$$
$$= 81$$

Therefore, the maximum value of $4(x + 7)(2 - x)$ over all real numbers $x$ is $\boxed{81}$.

Table 16: An example of the generated solution by RISE-QWEN2-7B.

---

**Problem 1**:

What is the maximum value of $4(x + 7)(2 - x)$, over all real numbers $x$?

**Reference Answer**:

81

---

**Qwen2-7B-Instruct**:

To find the maximum value of the function $f(x) = 4(x + 7)(2 - x)$, we can start by expanding the function and then completing the square or using calculus to find the critical points.

First, let's expand the function:

$$f(x) = 4(x + 7)(2 - x) = 4(\boxed{x^2 + 7x - 2x - 14}) = 4(x^2 + 5x - 14)$$

$$f(x) = 4x^2 + 20x - 56$$

Now, to find the maximum value, we can use calculus. We'll find the derivative of $f(x)$ with respect to $x$ and set it equal to zero to find the critical points.

$$f'(x) = \frac{d}{dx}(4x^2 + 20x - 56) = 8x + 20$$

Setting the derivative equal to zero to find the critical point:

$$8x + 20 = 0$$

$$8x = -20$$

$$x = -\frac{20}{8} = -\frac{5}{2}$$

Now, we need to determine if this critical point corresponds to a maximum. We can do this by checking the second derivative or by considering the nature of the parabola (since the coefficient of $x^2$ is positive, the parabola opens upwards, indicating that the critical point is a minimum, but we are looking for the maximum value of the function).

However, since we are looking for the maximum value of the function, and we've found that the function is a parabola opening upwards, the maximum value will occur at the endpoints of the domain or at the vertex of the parabola. Since the function is defined for all real numbers, we need to consider the behavior of the function as $x$ approaches positive and negative infinity.

Given the function is a parabola opening upwards, the maximum value will occur at the vertex. We've already found the x-coordinate of the vertex to be $x = -\frac{5}{2}$. To find the maximum value, we substitute this x-value back into the original function:

$$f\left(-\frac{5}{2}\right) = 4\left(-\frac{5}{2}\right)^2 + 20\left(-\frac{5}{2}\right) - 56$$

$$= 4\left(\frac{25}{4}\right) - 50 - 56$$

$$= 25 - 50 - 56$$

$$= -81$$

Therefore, the maximum value of $4(x + 7)(2 - x)$ over all real numbers $x$ is $-81$.

The answer is $\boxed{-81}$.

---

Table 17: An example of the generated solution by the general LLM Qwen2-7B-Instruct. The subtle error is put in a red box.

**Problem 2**:

If we write $\sqrt{2} + \frac{1}{\sqrt{2}} + \sqrt{3} + \frac{1}{\sqrt{3}}$ in the form $\dfrac{a\sqrt{2} + b\sqrt{3}}{c}$ such that $a$, $b$, and $c$ are positive integers and $c$ is as small as possible, then what is $a + b + c$?

**Reference Answer**:

23

**RISE-LLAMA-3.1-8B**:

To solve this problem, we need to rationalize the given expression and then simplify it.

Step 1: Rationalize the expression $\sqrt{2} + \frac{1}{\sqrt{2}} + \sqrt{3} + \frac{1}{\sqrt{3}}$.

To rationalize $\frac{1}{\sqrt{2}}$ and $\frac{1}{\sqrt{3}}$, we multiply each fraction by its conjugate over itself to eliminate the square root in the denominator.

$\frac{1}{\sqrt{2}} = \frac{1}{\sqrt{2}} \times \frac{\sqrt{2}}{\sqrt{2}} = \frac{\sqrt{2}}{2}$

$\frac{1}{\sqrt{3}} = \frac{1}{\sqrt{3}} \times \frac{\sqrt{3}}{\sqrt{3}} = \frac{\sqrt{3}}{3}$

Step 2: Substitute the rationalized fractions back into the original expression.

$\sqrt{2} + \frac{\sqrt{2}}{2} + \sqrt{3} + \frac{\sqrt{3}}{3}$

Step 3: Combine like terms.

$\sqrt{2} + \frac{\sqrt{2}}{2} = \frac{2\sqrt{2} + \sqrt{2}}{2} = \frac{3\sqrt{2}}{2}$

$\sqrt{3} + \frac{\sqrt{3}}{3} = \frac{3\sqrt{3} + \sqrt{3}}{3} = \frac{4\sqrt{3}}{3}$

So the expression becomes $\frac{3\sqrt{2}}{2} + \frac{4\sqrt{3}}{3}$.

Step 4: To combine these two fractions into one fraction with a common denominator, we need to find the least common multiple (LCM) of 2 and 3, which is 6.

$\frac{3\sqrt{2}}{2} = \frac{3\sqrt{2} \times 3}{2 \times 3} = \frac{9\sqrt{2}}{6}$

$\frac{4\sqrt{3}}{3} = \frac{4\sqrt{3} \times 2}{3 \times 2} = \frac{8\sqrt{3}}{6}$

Step 5: Combine the two fractions.

$\frac{9\sqrt{2}}{6} + \frac{8\sqrt{3}}{6} = \frac{9\sqrt{2} + 8\sqrt{3}}{6}$

Now we have the expression in the desired form $\dfrac{a\sqrt{2} + b\sqrt{3}}{c}$.

Step 6: Identify the values of a, b, and c.

$a = 9$, $b = 8$, and $c = 6$.

Step 7: Calculate the sum of a, b, and c.

$a + b + c = 9 + 8 + 6 = 23$

The final answer is: $\boxed{23}$

Table 18: An example of the generated solution by RISE-LLAMA-3.1-8B.

**Problem 2**:

If we write $\sqrt{2} + \frac{1}{\sqrt{2}} + \sqrt{3} + \frac{1}{\sqrt{3}}$ in the form $\dfrac{a\sqrt{2} + b\sqrt{3}}{c}$ such that $a$, $b$, and $c$ are positive integers and $c$ is as small as possible, then what is $a + b + c$?

**Reference Answer**:

23

**Llama-3.1-8B-Instruct**:

To simplify the given expression, we can start by rationalizing the denominators of the fractions.

$\sqrt{2} + \frac{1}{\sqrt{2}} + \sqrt{3} + \frac{1}{\sqrt{3}}$

First, we rationalize the denominators of the fractions by multiplying each fraction by the conjugate of its denominator over itself.

$\sqrt{2} + \frac{1}{\sqrt{2}} + \sqrt{3} + \frac{1}{\sqrt{3}} = \sqrt{2} + \frac{\sqrt{2}}{2} + \sqrt{3} + \frac{\sqrt{3}}{3}$

Now, we can combine like terms.

$\sqrt{2} + \boxed{\dfrac{\sqrt{2}}{2}} + \sqrt{3} + \frac{\sqrt{3}}{3} = \frac{4\sqrt{2}}{2} + \boxed{\dfrac{2\sqrt{2}}{2}} + \frac{3\sqrt{3}}{3} + \frac{\sqrt{3}}{3}$

Combine the like terms.

$\frac{4\sqrt{2}}{2} + \frac{2\sqrt{2}}{2} + \frac{3\sqrt{3}}{3} + \frac{\sqrt{3}}{3} = \frac{6\sqrt{2}}{2} + \frac{4\sqrt{3}}{3}$

Simplify the fractions.

$\frac{6\sqrt{2}}{2} + \frac{4\sqrt{3}}{3} = 3\sqrt{2} + \frac{4\sqrt{3}}{3}$

To get a common denominator, we multiply the first term by $\frac{3}{3}$.

$3\sqrt{2} + \frac{4\sqrt{3}}{3} = \frac{9\sqrt{2}}{3} + \frac{4\sqrt{3}}{3}$

Now we can combine the fractions.

$\frac{9\sqrt{2}}{3} + \frac{4\sqrt{3}}{3} = \frac{9\sqrt{2} + 4\sqrt{3}}{3}$

Now we have the expression in the desired form $\dfrac{a\sqrt{2} + b\sqrt{3}}{c}$, where $a = 9$, $b = 4$, and $c = 3$.

Therefore, $a + b + c = 9 + 4 + 3 = 16$.

The final answer is: $\boxed{16}$

Table 19: An example of the generated solution by the general LLM Llama-3.1-8B-Instruct. The subtle error is put in a red box.

**Prompt for Numerical Calculation Errors**:

Question:
{question}

Initial Answer:
{answer}

Current Step:
{text}

Edit a numerical value or a series of related values in the current step to make a wrong calculation. Do not state that errors have been made.

Table 20: The prompt for injecting Numerical Calculation Errors.

**Prompt for Numeric or Symbolic Substitution Errors**:

Question:
{question}

Initial Answer:
{answer}

Current Step:
{text}

Edit a value or symbol in the current step to make a wrong substitution. Do not state that errors have been made.

Table 21: The prompt for injecting Numeric or Symbolic Substitution Errors

**Prompt for Omission of Calculation Terms**:

Question:
{question}

Initial Answer:
{answer}

Current Step:
{text}

Delete a calculation term in the current step to make a wrong calculation. Do not state that errors have been made.

Table 22: The prompt for injecting Omission of Calculation Terms.

**Prompt for Errors in the Calculation Order**:

Question:
{question}

Initial Answer:
{answer}

Current Step:
{text}

Swap two calculation terms in the current step to make a wrong calculation. Do not state that errors have been made.

Table 23: The prompt for injecting Errors in the Calculation Order.

**Prompt for Errors in the Use of Calculation Symbols**:

Question:
{question}

Initial Answer:
{answer}

Current Step:
{text}

Edit a calculation symbol (e.g., +-*/, etc.) in the current step to make a wrong calculation.

Table 24: The prompt for injecting Errors in the Use of Calculation Symbols.

