# OpenReview forum: "Subtle Errors Matter: Preference Learning via Error-injected Self-editing"
_ICLR.cc/2025/Conference — Submitted to ICLR 2025_

### Official Review · Reviewer_r3F2 · 2024-10-27

**Soundness:** 3
**Presentation:** 3
**Contribution:** 3
**Rating:** 6
**Confidence:** 3

**Summary:**

This paper proposes a novel approach to enhancing the performance of language models in mathematical problem-solving by introducing noise into correct solutions to create subtly incorrect answers. These correct/incorrect answer pairs are then utilized to fine-tune a DPO (Direct Preference Optimization) model. Empirical evidence demonstrates the effectiveness of this approach in improving the model’s accuracy.

**Strengths:**

1. The paper effectively addresses the challenge of subtle error detection, a common issue faced by large language models (LLMs). The proposed method successfully enhances models, transforming relatively weaker ones into stronger versions.
1. The technique leverages the simplicity of generating incorrect solutions rather than correct ones, making the training process more efficient. By instructing models to produce errors, it harnesses the LLM’s capability to identify likely mistakes based on its own tendencies, leading to a self-improvement mechanism.

**Weaknesses:**

1. The effectiveness of this method heavily relies on prompt engineering. The quality and specificity of the prompts used to generate incorrect answers significantly influence the quality and subtlety of the generated mistakes.
1. As the approach is based on pre-defined templates, it may not capture the full spectrum of potential errors a language model might make, leaving certain blind spots unaddressed.
1. The scalability of the proposed method remains uncertain, as it primarily focuses on generating diverse incorrect answers rather than ensuring diversity in correct solutions. This might limit its applicability in more complex or varied scenarios.

**Questions:**

1. Do you think this method can be extended to other domains where subtle mistake detection is crucial, such as logical reasoning and programming? If so, what adaptations would be necessary?

---

> ### Author Response · Authors · 2024-11-21
> **Response to Official Review by Reviewer r3F2 (1/3)**
>
> Thanks a lot for your thoughtful and positive review. We appreciate your suggestions and would like to address the concerns you raised:
>
> **1. Reliance on Prompt Engineering**
>
> We acknowledge that prompt engineering helps guide the generation of specific types of incorrect answers. However, it is important to note that the design of prompts can be flexible and adaptable. To reduce reliance on manual prompt engineering and demonstrate the flexibility of prompts used in RISE, we use the self-instruct method to generate a variety of prompt templates (10 templates for each type of error) and conduct self-editing with a random choice of the generated prompts. Some examples of prompt templates are shown as follows:
>
> **REPLACE a numerical value:**
>
> (1) Change a number in this step so that the calculation becomes incorrect, without indicating that a mistake has been introduced.
>
> (2) Alter the numerical value in this stage to produce an incorrect result, but avoid mentioning the error.
>
> (3) Modify a number in the current calculation to lead to a wrong outcome, without revealing the inaccuracy.
>
> (4) Adjust one of the values in this step to ensure the calculation is wrong, without pointing out the error.
>
> (5) Replace a number in the calculation with an incorrect one, but do not mention that anything is wrong.
>
> (6) Change a figure at this point to cause an erroneous result, without disclosing that you've made a mistake.
>
> (7) Introduce a wrong number in this calculation step, but refrain from stating that an error has occurred.
>
> (8) Modify a numerical value here so that the result is incorrect, without drawing attention to the mistake.
>
> (9) Adjust the number in this step to generate an inaccurate result, without acknowledging the error.
>
> (10) Introduce an incorrect value in this calculation, but avoid mentioning that the outcome is wrong.
>
> **SWAP two calculation terms:**
>
> (1) Switch the positions of two terms in the current calculation step to lead to an incorrect result, without explicitly acknowledging the mistake.
>
> (2) Rearrange two terms in the present step in a way that causes an error, but avoid mentioning that a mistake has occurred.
>
> (3) Alter the order of two terms in the current calculation to produce an incorrect outcome, without pointing out the error.
>
> (4) Exchange the positions of two terms in this step to intentionally create a miscalculation, and don't indicate that anything is wrong.
>
> (5) Adjust the placement of two terms in the ongoing calculation to introduce an error, without drawing attention to the fact.
>
> (6) Swap the order of two terms in the current process to result in a wrong answer, but refrain from noting the mistake.
>
> (7) Change the arrangement of two terms in the current step in a way that leads to an incorrect result, without signaling any error.
>
> (8) Interchange two terms in the current calculation step to produce a mistake, while keeping the error implicit.
>
> (9) Shift the positions of two terms in the calculation to create a wrong result, without stating that something is incorrect.
>
> (10) Modify the sequence of two terms in this step, causing an incorrect calculation, but don't mention the flaw.
>
> Preliminary results on Qwen2-7B-Instruct with the above self-edited samples are shown as follows:
>
> | Method | GSM8K | MATH |
> | :-----| :----: | :----: |
> | RISE-prompt-manual-error | 88.40 | 59.90 |
> | RISE-prompt-self-instruct-error | 88.55 | 59.32 |
>
> With a random selection of prompt templates, our RISE can still help improve mathematical reasoning capability and outperform the general DPO method. Compared with the results of the manual prompts used in our paper, the results of self-instruct prompts show a better accuracy on GSM8K but a slightly worse accuracy on MATH.

---

> ### Author Response · Authors · 2024-11-21
> **Response to Official Review by Reviewer r3F2 (2/3)**
>
> **2. Capturing the Full Spectrum of Potential Errors**
>
> We agree that pre-defined templates may have limitations in capturing the full spectrum of potential errors. To further illustrate that our approach has the potential to be generalized to more diverse errors, we implement another experiment with a more universal prompt template. The prompt template is as follows:
>
> "*Edit the current step to introduce an error. Do not state that errors have been made.*"
>
> This prompt doesn’t indicate any error types and leverages the LLM itself to randomly introduce an error, which can capture broader spectrum error types. More importantly, this prompt can introduce arbitrary errors and even unexposed errors. Preliminary results on Qwen2-7B-Instruct with these self-edited samples are shown as follows:
>
> |Method | GSM8K | MATH|
> | :-----| :----: | :----: |
> |RISE-prompt-pre-defined-error |88.4| 59.9|
> |RISE-prompt-arbitrary-error|88.3|59.7|
>
> The results show a similar significant improvement compared with the results on our pre-defined prompt templates.
>
> Besides, we consider creating prompt templates with more error types by comprehensive GPT-4o-based error analysis. We can use the self-instruct method with the analyzed errors to create prompt templates automatically. However, our experiments have already demonstrated the feasibility of this framework, as the error types used in our experiments are systematically identified and summarized through GPT-4o-based error analysis.

---

> ### Author Response · Authors · 2024-11-22
> **Response to Official Review by Reviewer r3F2 (3/3)**
>
> **3. Scalability of the Method & Applicability to Other Domains**
>
> We appreciate this observation and agree that scalability is an important consideration. A key algorithmic feature of our method, as well as other DPO-like methods, is the in-distribution sampling of chosen or rejected solutions from the policy model's learned distribution, followed by targeted in-distribution optimization to better align the model's responses with human preferences. Consequently, these methods may not have originally been designed to improve response diversity. Our method, in particular, primarily enhances the model's ability to minimize subtle errors, enabling it to consistently arrive at correct solutions with greater reliability.
>
> We believe that our method can be applied in more complex or varied scenarios and be extended to other domains, such as logical reasoning and programming, where subtle mistake detection is essential. Given the results of self-editing with an extremely simple prompt (“introduce an error”), we seamlessly implement our RISE on the code generation task. The prompt is the same as above:
>
> "*Edit the current step to introduce an error. Do not state that errors have been made.*"
>
> This prompt can introduce arbitrary errors and can be adapted to other domains such as code generation easily. I am still processing the code dataset, and the detailed results will be released soon.

---

> > ### Author Response · Authors · 2024-11-24
> > **3. Scalability of the Method & Applicability to Other Domains**
> >
> > We apply our RISE method to code generation, where avoiding subtle errors is critical. Following [1], we adopt the LeetCode dataset [2] to conduct training. The dataset includes around 2K leetcode tasks in the medium and hard levels. For the Qwen2-7B-Instruct model, we sample 50 times and finally obtain 1473 pairs of chosen and rejected samples for training. The preliminary results are shown as follows:
> >
> > | Method | MBPP | Humaneval |
> > | :-----| :----: | :----: |
> > | Qwen2-7B-Instruct | 42.2 | 43.9 |
> > | Qwen2-7B-DPO | 43.4 | 46.3 |
> > | Qwen2-7B-RISE | **44.2** | **47.6** |
> >
> > We can observe that our RISE performs better than the general DPO method, achieving a 0.8% improvement on the MBPP test set and a 1.3% improvement on the Humaneval test set. Considering that the current solution editing strategy has not yet been adjusted to account for the characteristics of code generation, there is still certain room for improvement in the results.
> >
> > [1] Xu, Bin, Yiguan Lin, and Yinghao Li. "SRA-MCTS: Self-driven Reasoning Aurmentation with Monte Carlo Tree Search for Enhanced Code Generation." arXiv preprint arXiv:2411.11053 (2024).
> >
> > [2] https://huggingface.co/datasets/greengerong/leetcode

---

> ### Author Response · Authors · 2024-11-25
> **Request for Feedback on Our Rebuttal**
>
> Dear Reviewer r3F2,
> \
> \
> We hope this email finds you well. Thank you again for your thoughtful review and constructive feedback on our submission.
>
> As the rebuttal period is approaching its end, we wanted to kindly follow up to request your feedback on our responses. We have worked hard to address your concerns through detailed explanations and additional experiments, and we would greatly appreciate hearing your thoughts on whether our rebuttal has adequately addressed your questions.
>
> If you have any remaining concerns or need further clarification, we would be happy to provide additional details before the rebuttal deadline. Thank you again for your time and invaluable contributions to improving our work.
> \
> \
> Sincerely,
>
> Authors of Paper #10957

---

> > ### Comment · Area_Chair_t9mg · 2024-11-26
> >
> > Dear reviewer r3F2,
> >
> > Could you please respond to authors' rebuttal and see if you would like to update your review? Thanks very much!
> >
> > AC

---

> ### Author Response · Authors · 2024-12-02
> **Request for Feedback on Our Rebuttal**
>
> Dear Reviewer r3F2,
> \
> \
> I hope this email finds you well. As the rebuttal period is approaching its conclusion, we wanted to kindly follow up regarding your feedback on our submission. Your insights are incredibly important to us, and we would greatly appreciate it if you could share your response at your earliest convenience.
>
> We have carefully addressed the points raised in the initial reviews and conducted additional experiments to strengthen our submission. We believe these improvements significantly enhance the quality of the work, and we hope you might take them into consideration when reassessing the paper and its score.
>
> Thank you very much for your time, effort, and thoughtful evaluation. Please let us know if there are any further concerns or questions we can address to assist you.
> \
> \
> Authors of Paper #10957

---

> ### Author Response · Authors · 2024-12-03
> **Request for Feedback on Our Rebuttal**
>
> Dear Reviewer r3F2,
> \
> \
> The rebuttal period is ending in just a few hours, and we kindly remind you to provide your feedback. We have thoroughly addressed your concerns and conducted additional experiments to strengthen our submission. We sincerely hope you will consider these improvements when reassessing the paper and its score.
>
> Thank you very much for your time and effort!
> \
> \
> Authors of Paper #10957

---

### Official Review · Reviewer_nbYg · 2024-11-02

**Soundness:** 3
**Presentation:** 3
**Contribution:** 3
**Rating:** 5
**Confidence:** 2

**Summary:**

This paper presents a model fine-tuning method that aims to solve the math problem of LLMs. Specifically, the paper reuses the existing training paradigm (RLFH or DPO concretely) but pays attention to the training pair generation. The negative sample is generated by prompting LLM with intentional instruction on producing wrong answers (with particular error types described in Section 2.1). The main motivation comes from the hypothesis that the existing fine-tuning solution does not provide targetted training pairs, causing the fine-tuned model to capture subtle errors that are not intended.

The training objectives are borrowed from previous works, including DPO, step-wise DPO, and negative log-likelihood loss (to stabilize the training). The contribution of this paper is more about how to get better negative samples. Experiments show the proposed approach provides reasonable improvement on small models (7B) but with limited improvement on large models (70B). Compared to the DPO solution previously used, the improvement is also not very significant.

**Strengths:**

The paper is well-written and easy to understand. The majority of the descriptions are with clear equations or demonstrative figure supports. While some contents are not included in the paper, they are well-known in the literature.

Actively looking for hard negative examples is always a research topic. The paper focuses on generating regularized negatives (from the list of error types) such as enforcing the fine-tuned model to avoid making similar mistakes. The strategy is letting LLM provide the wrong examples as part of forward passing over steps.

Experiments included multiple LLMs in comparison. While most of them are not intended to solve math problems, it is good to have more things to compare.

**Weaknesses:**

The most obvious weakness of this paper is the lack of solid evidence of motivation. While the paper stated that negative samples randomly generated may be unrelated to error, there is no solid evidence on how it impacts the model's math ability. The research gap has yet to be explicitly visible so far. The authors should provide sufficient insight into how those randomly sampled negatives hurt the performance with solid evidence. I am not very convinced this issue is significant. Considering the limited performance improvement on large models (70B), I am concerned if this is indeed a problem of existing works.

Most of the approaches used in this paper are existing training methods. The novelty of this paper is weak, given it is simply proposing a negative sample generation approach. The majority of the training descriptions are optional, given they are well-known. E.g. DPO, step-wise DPO, or NLL. The authors give good justification for why using prompts to generate the wrong sample is an impactful contribution.

The last point of concern is the performance improvement of large models. The proposed method does not seem to improve the performance of the large models much. Why is that? It seems not an improvement space problem since the large LLMs still suffer from math problems with many errors. However, even after this fine-tuning, the proposed solution fails to take care of them. Any insight? Is it indicating the proposed solution is going in the wrong direction for solving this problem?

**Questions:**

1. Why does this solution fail to improve the math of large models compared to the models fine-tuned on random negative samples? If someone uses a very simple negative sampling approach by generating random numerical values, is there any performance difference from the proposed solution?

---

> ### Author Response · Authors · 2024-11-22
> **Response to Official Review by Reviewer nbYg (1/3)**
>
> Thank you for your detailed review and constructive feedback. We appreciate your valuable comments and would like to address the concerns you raised:
>
> **1. Motivation and Impact of Random Rejected Samples**
>
> We would like to clarify that we do not claim that using randomly sampled negative solutions (rejected samples) harms the model's mathematical reasoning ability. On the contrary, our ablation experiments (- w/o self-edited pairs: GSM8K: 88.3 (+2.9), MATH: 58.2 (+6.0)) demonstrate that even using only randomly sampled negative solutions can help the model achieve correct solutions more consistently. Randomly sampled rejected samples are effective for preference learning since they do involve wrong answers. Furthermore, our method constructs self-edited pairs and supplements them into the existing randomly sampled rejected samples. These pairs, as finer-grained preference pairs, further enhance the model's ability to reduce small errors (RISE-QWEN2-7B: GSM8K: 88.4 (+3.0), MATH: 59.9 (+7.7)).
>
> Our motivation is that randomly sampled rejected samples make it difficult for preference learning to focus on subtle errors, as the differences between chosen and rejected samples often include content unrelated to errors. Our self-edited pairs can make up for this deficiency. A randomly sampled pair is shown below:
>
> **Chosen:**
>
> "Let's think step by step.
>
> Step 1:  Recognize that if $-1 - 4\sqrt{2}$ is a root, then its conjugate $-1 + 4\sqrt{2}$ must also be a root because the coefficients of the polynomial are rational.
>
> Step 2:  We can use Vieta's formulas, which relate the coefficients of a polynomial to sums and products of its roots. For a cubic polynomial $x^3 + ax^2 + bx + c = 0$, the sum of the roots is $-a$, the sum of the products of the roots taken two at a time is $b$, and the product of the roots is $-c$.
>
> Step 3:  Let's denote the roots as $r_1 = -1 - 4\sqrt{2}$, $r_2 = -1 + 4\sqrt{2}$, and $r_3$. Since we know the product of the roots is $-31$, we have:
> \[r_1 \cdot r_2 \cdot r_3 = -31\]
>
> Step 4:  Calculate the product of the first two roots:
> \[r_1 \cdot r_2 = (-1 - 4\sqrt{2})(-1 + 4\sqrt{2}) = 1 - (4\sqrt{2})^2 = 1 - 32 = -31\]
>
> Step 5:  Knowing that the product of the roots is $-31$, and the product of the first two roots is $-31$, we can deduce that $r_3 = 1$.
>
> Step 6:  Now, calculate the sum of the roots:
> \[r_1 + r_2 + r_3 = (-1 - 4\sqrt{2}) + (-1 + 4\sqrt{2}) + 1 = -1 - 1 + 1 = -1\]
>
> Step 7:  According to Vieta's formulas, the sum of the roots is $-a$. Therefore:
> \[-a = -1 \Rightarrow a = 1\]
>
> So, the value of $a$ is $\boxed{1}$."
>
> **Rejected:**
>
> "Let's think step by step.
>
> Step 1:  Since the coefficients of the polynomial are rational, and the polynomial has rational coefficients, the complex conjugate of $-1 - 4\sqrt{2}$ must also be a root of the polynomial. The complex conjugate of $-1 - 4\sqrt{2}$ is $-1 + 4\sqrt{2}$. Therefore, the polynomial has the form \[(x + 1 + 4\sqrt{2})(x + 1 - 4\sqrt{2})(x - r) = 0\] for some rational number $r$.
>
> Step 2:  We can multiply the first two factors to get \[(x^2 + 2x + 17)(x - r) = 0.\] Expanding this product, we get \[x^3 + (2 - r)x^2 + (17 - 2r)x - 17r = 0.\] Comparing this with the given polynomial, we see that \[a = 2 - r, \quad b = 17 - 2r, \quad \text{and} \quad -17r = 31.\] From the last equation, we get $r = -\frac{31}{17} = -\frac{31}{17}$.
>
> Step 3:  Substituting $r = -\frac{31}{17}$ into the equation $a = 2 - r$, we get \[a = 2 - \left(-\frac{31}{17}\right) = 2 + \frac{31}{17} = \frac{34}{17} + \frac{31}{17} = \frac{65}{17}.\] Therefore, the value of $a$ is $\boxed{\frac{65}{17}}$. The answer is: $\boxed{\frac{65}{17}}$"
>
> From the above two samples, it is hard to directly locate errors in the rejected by comparing the chosen and rejected samples token by token, as there is a significant difference in the expression of these two samples. However, our self-edited pairs can help the model focus on subtle errors more conveniently. A self-edited pair is shown below:
>
> **Chosen:**
>
> "Let's think step by step.
>
> Step 1:  Determine the total hours one pencil can last.
> Since Jenine can sharpen a pencil 5 times before it becomes unusable and she needs to sharpen a pencil every 1.5 hours of use, one pencil can last for \(5 \times 1.5 = 7.5\) hours."
>
> **Rejected:**
>
> "Let's think step by step.
>
> Step 1:  Determine the total hours one pencil can last.
> Since Jenine can sharpen a pencil 5 times before it becomes unusable and she needs to sharpen a pencil every 1.5 hours of use, one pencil can last for \(5 - 1.5 = 3.5\) hours."
>
> From the above two samples, we can observe that it is easier for the model to focus on the subtle but key errors in the generated samples, as there is only a symbol difference ("\times" <-> "-") between them. Thus, we believe that our motivation is reasonable, as demonstrated by our comprehensive experiments.

---

> ### Author Response · Authors · 2024-11-22
> **Response to Official Review by Reviewer nbYg (2/3)**
>
> **2. Novelty of the Approach**
>
> We understand the concern regarding the novelty of the approach. We design a simple and effective framework to improve the LLM capability to avoid subtle errors. The novelty of the approach lies more in the use of **LLM itself** to generate **subtle, domain-specific incorrect solutions** that are crucial and effective for improving model robustness. Instead of sampling preference pairs, we utilize editing approaches to generate rejected samples, which is more efficient for models to learn to avoid specific subtle errors. Moreover, subtle errors are an important but often overlooked issue.
>
> In addition, constructing more effective preference data is becoming increasingly important for preference learning, especially as human preferences may fail in more complex scenarios. Our method uses the model itself to generate fine-grained preference pairs and focus on subtle but essential errors. What sets this work apart is the focus on **leveraging the model’s own understanding** to make realistic errors, which helps it learn to avoid similar mistakes. Thus, our method has the potential to enable models to **self-improve**.

---

> ### Author Response · Authors · 2024-11-22
> **Response to Official Review by Reviewer nbYg (3/3)**
>
> **3. Performance Improvement on Large Models**
>
> We appreciate the reviewer’s observation. The reduced performance gains on large models is indeed an interesting result. Our hypothesis is that larger models, while still prone to errors, may have already learned more sophisticated representations of mathematical reasoning during pre-training. As a result, the room for improvement through preference learning is smaller compared to smaller models. Similar phenomenon can be found in DART-math [1], where fine-tuning the Llama3-70B model achieves a small accuracy gain on most in-domain and out-of-domain mathematical evaluation datasets. Some metrics even decrease after further fine-tuning.
>
> In our experiments, we adopt mathematical problems in commonly recognized and used datasets such as MetaMATH and AQuA and finally collect 4K pairs for preference learning of large models, corresponding to only 2K problems. Even if the sampling attempt  is increased, the number of effective pairs that can be collected does not increase significantly. This number of problems may be inadequate to help large models further improve a lot.
>
> [1] Tong, Yuxuan, et al. "Dart-math: Difficulty-aware rejection tuning for mathematical problem-solving." arXiv preprint arXiv:2407.13690 (2024).

---

> ### Author Response · Authors · 2024-11-25
> **Request for Feedback on Our Rebuttal**
>
> Dear Reviewer nbYg,
> \
> \
> We hope this message finds you well. Thank you again for your thoughtful feedback on our submission.
>
> As the rebuttal period is coming to a close, we wanted to kindly follow up to request your feedback on our responses. We have made every effort to address your concerns through detailed clarifications, and we would greatly appreciate hearing whether our rebuttal has resolved your questions.
>
> If you have any remaining concerns or require further clarification, we would be happy to address them before the rebuttal deadline. Thank you again for your time and effort in reviewing our work.
> \
> \
> Sincerely,
>
> Authors of Paper #10957

---

> > ### Comment · Reviewer_nbYg · 2024-11-26
> >
> > Thank you for your efforts in preparing the rebuttal. I have reviewed your response and decided to maintain my current score for two main reasons: (1) concerns about the novelty of the work and (2) the performance limitations with the larger model, which reduce the practical utility of the study. While the authors have provided a thorough and well-prepared rebuttal, the limitations of the work remains (nothing about the clarity of presentation). That said, I would not oppose acceptance if other reviewers advocate for it.

---

> > > ### Author Response · Authors · 2024-11-27
> > > **Response to Reviewer nbYg**
> > >
> > > Thank you very much for your kind reply! We are currently conducting experiments on Llama-3.1-70B-Instruct and Qwen2-72B-Instruct again. In comparison to the 7B-level experiments (k=5), we are significantly increasing the number of sampling attempts (k=50) to generate pairs for a larger number of problems. We hope these experiments will address your concerns about the performance limitations of the larger model.
> > >
> > > Due to equipment limitations, these experiments may require some additional time, but we will share the results as soon as they are available.
> > > \
> > > \
> > > Sincerely,
> > >
> > > Authors of Paper #10957

---

> > > ### Author Response · Authors · 2024-12-02
> > > **Comments on performance with larger models**
> > >
> > > Dear Reviewer nbYg,
> > > \
> > > \
> > > We have re-implemented experiments on Llama-3.1-70B-Instruct, increasing the number of sampling attempts to obtain more preference sample pairs for training. By conducting a direct comparison between our proposed method RISE and the baseline DPO, we demonstrate the clear advantages of RISE in multiple math test datasets. The preliminary results are summarized in the table below:
> > >
> > > | Method | GSM8K | MATH | AQuA |SVAMP | AIME24 | Odyssey-MATH |
> > > | :-----| :----: | :----: | :----: | :----: | :----: | :----: |
> > > | Llama-3.1-70B-Instruct |  94.9 | 65.0|77.1|93.0|7/30|**60.4**|
> > > | DPO | 94.5|65.5|77.1|93.1|7/30|58.4|
> > > | RISE | **95.2**|**66.4**|**78.7**|**93.5**|7/30|60.0|
> > >
> > > On MATH, RISE achieves a score of 66.4, compared to 65.5 for DPO and 65.0 for the base model. Similarly, on AQuA, RISE scores 78.7, outperforming both DPO (77.1) and the base model (77.1). These results highlight RISE's ability to address complex mathematical and logical reasoning tasks with greater accuracy and consistency.
> > >
> > > On GSM8K, RISE achieves the highest score of 95.2, surpassing DPO (94.5) and the base model (94.9). For SVAMP, RISE improves slightly over DPO (93.5 vs. 93.1) and the base model (93.0), showcasing its reliability in solving various types of problems.
> > >
> > > For the more challenging datasets, Odyssey-MATH, RISE remains competitive. RISE can maintain reasoning capability for complex tasks while DPO slightly hurts performance. These findings underscore RISE’s robustness and effectiveness as a method for improving reasoning in larger language models. **Moreover, it performs better than the original DPO method.**
> > > \
> > > \
> > > Authors of Paper #10957

---

> > > ### Author Response · Authors · 2024-12-03
> > > **Request for Feedback on Our Rebuttal**
> > >
> > > Dear Reviewer nbYg,
> > > \
> > > \
> > > With only a few hours remaining in the rebuttal period, we kindly remind you to provide your feedback at your earliest convenience. We have carefully addressed your concerns and added new experiments to strengthen the submission. We sincerely hope you will consider these improvements when revisiting the paper and its score.
> > >
> > > Thank you so much for your time and consideration!
> > > \
> > > \
> > > Authors of Paper #10957

---

### Official Review · Reviewer_kiSm · 2024-11-03

**Soundness:** 3
**Presentation:** 3
**Contribution:** 3
**Rating:** 6
**Confidence:** 4

**Summary:**

The paper presents a novel preference learning framework known as eRror-Injected Self-Editing (RISE), which is designed to enhance the mathematical reasoning capabilities of Large Language Models (LLMs). The core contribution of RISE is its innovative approach to error mitigation by injecting subtle, predefined errors into correct solutions, creating hard pairs for training that help the model focus on common mistake patterns.
The framework operates by using the LLM to generate correct multi-step solutions and then intentionally introducing errors into these solutions to form self-edited pairs. These pairs, along with correctly and incorrectly solved samples, are used for subtle error-aware Direct Preference Optimization (DPO) training. The paper reports improvements on mathematical reasoning tasks when RISE is applied to LLMs like Qwen2-7B-Instruct, and shows improvement on the GSM8K and MATH dataset. These results demonstrate the effectiveness of RISE in refining the training objective to target subtle error tokens and in improving the model's ability.

**Strengths:**

1. Originality: The paper introduces RISE, a novel preference learning framework that addresses the subtle error problem in LLMs, which is not commonly seen in other works.
2. Quality: The paper compares RISE with other preference learning methods and shows that it outperforms them, which shows the quality of the approach.
3. Significance: The paper makes a significant contribution to the field of LLMs by providing a method to improve their mathematical reasoning capabilities, which is a critical area for LLM development.

**Weaknesses:**

1. Generalizability and Domain-Specificity: The paper focuses exclusively on mathematical reasoning tasks. It would be beneficial to see how RISE performs in other domains, such as reasoning and coding, where subtle errors also play a significant role.
2. Diversity of Error Types: While the paper addresses common error types in mathematical reasoning, it may not cover all possible error categories. For instance, it might be worth exploring errors related to the interaction of different error types.
3. Experiments: The authors should evaluate the performance of RICE on more open-source models (such as Mistral-7B-Instruct-v0.3, qwen 2.5 series). And the influence of hyperparameter $\alpha$ should also be explored.

**Questions:**

1. The interaction of different error types: Will the interaction of different error types improve the performance?
2. Number of self-edited pairs: Why the performance on GSM8K and MATH both decrease with more self-edited pairs? Please further analyse this pheonomenon to pre-define the number of self-edited pairs.

---

> ### Author Response · Authors · 2024-11-21
> **Response to Official Review by Reviewer kiSm (1/4)**
>
> Thank you for your detailed review and thoughtful feedback. We appreciate your recognition of our work and would like to address the concerns and questions you raised:
>
> **1. Generalizability and Domain-Specificity**
>
> We agree that expanding the evaluation to additional domains would provide a more comprehensive understanding of RISE's generalizability. We apply our RISE to other domains such as code generation, where subtle mistake detection is essential. The editing prompt is as above:
>
> "*Edit the current step to introduce an error. Do not state that errors have been made.*"
>
> This prompt can introduce arbitrary errors and can be adapted to other domains easily. I am still processing the code dataset, and the detailed results will be released soon. Other relevant experiments that use such a universal prompt on our mathematical dataset have been conducted, and the results demonstrate the effectiveness of our RISE (refer to the (2/4) response).

---

> > ### Author Response · Authors · 2024-11-24
> > **1. Generalizability and Domain-Specificity**
> >
> > We apply our RISE method to code generation, where avoiding subtle errors is critical. Following [1], we adopt the LeetCode dataset [2] to conduct training. The dataset includes around 2K leetcode tasks in the medium and hard levels. For the Qwen2-7B-Instruct model, we sample 50 times and finally obtain 1473 pairs of chosen and rejected samples for training. The preliminary results are shown as follows:
> >
> > | Method | MBPP | Humaneval |
> > | :-----| :----: | :----: |
> > | Qwen2-7B-Instruct | 42.2 | 43.9 |
> > | Qwen2-7B-DPO | 43.4 | 46.3 |
> > | Qwen2-7B-RISE | **44.2** | **47.6** |
> >
> > We can observe that our RISE performs better than the general DPO method, achieving a 0.8% improvement on the MBPP test set and a 1.3% improvement on the Humaneval test set. Considering that the current solution editing strategy has not yet been adjusted to account for the characteristics of code generation, there is still certain room for improvement in the results.
> >
> > [1] Xu, Bin, Yiguan Lin, and Yinghao Li. "SRA-MCTS: Self-driven Reasoning Aurmentation with Monte Carlo Tree Search for Enhanced Code Generation." arXiv preprint arXiv:2411.11053 (2024).
> >
> > [2] https://huggingface.co/datasets/greengerong/leetcode

---

> ### Author Response · Authors · 2024-11-21
> **Response to Official Review by Reviewer kiSm (2/4)**
>
> **2. Diversity of Error Types & Interaction of Different Error Types**
>
> To illustrate that our approach has the potential to be generalized to more diverse errors, we implement another experiment with a more universal prompt template. The prompt template is as follows:
>
> "*Edit the current step to introduce an error. Do not state that errors have been made.*"
>
> This prompt doesn’t indicate any error types and leverages the LLM itself to randomly introduce an error, which can capture broader spectrum error types. More importantly, this prompt can introduce arbitrary errors. Preliminary results on Qwen2-7B-Instruct with these self-edited samples are shown as follows:
>
> | Method | GSM8K | MATH |
> | :-----| :----: | :----: |
> | RISE-prompt-pre-defined-error | 88.4 | 59.9 |
> | RISE-prompt-arbitrary-error | 88.3 | 59.7 |
>
> The results show a similar significant improvement compared with the results on our pre-defined prompt templates. However, as our injected errors are based on comprehensive error analysis and are better aligned with the practical situation, the performance with pre-defined error prompts is slightly better than that with arbitrary error.
>
> As for the interaction of different error types, we believe that the interaction of different error types could indeed improve performance by making the training data more diverse and challenging, potentially helping models learn to generalize better. We create a prompt with the interaction of different errors and conduct another experiment. Preliminary results on Qwen2-7B-Instruct with these self-edited samples are shown as follows:
>
> | Method | GSM8K | MATH |
> | :-----| :----: | :----: |
> | RISE-prompt-single-error | 88.4 | 59.9 |
> | RISE-prompt-combination-error | 88.7 | 60.0 |
>
> We can see that these compound error-injected samples help the model further improve the mathematical performance.

---

> ### Author Response · Authors · 2024-11-21
> **Response to Official Review by Reviewer kiSm (3/4)**
>
> **3. Experiments on More Open-Source Models**
>
> We appreciate this suggestion and agree that evaluating RISE on a broader range of open-source models could further validate its effectiveness. We implement additional experiments on Ministral-8B-Instruct and Qwen2.5-7B-Instruct, as these models are the most recent and well-regarded for their performance in various reasoning tasks. Preliminary results on these models show the effectiveness of our framework as follows:
>
> | Method | GSM8K | MATH |
> | :-----| :----: | :----: |
> | Ministral-8B-Instruct-2410 | 86.35 | 53.62 |
> | Ministral-8B-DPO | 86.95 | 54.18 |
> | Ministral-8B-RISE | **88.62** | **54.86** |
>
> | Method | GSM8K | MATH |
> | :-----| :----: | :----: |
> |Qwen2.5-7b-Instruct| 91.81 | 74.36 |
> |Qwen2.5-7b-DPO| 92.49 | 75.00 |
> |Qwen2.5-7b-RISE| **92.95** | **75.06** |
>
> We can observe that RISE significantly improves the mathematical performance of both Ministral-8B-Instruct-2410 and Qwen2.5-7b-instruct. Especially for the Ministral model, the accuracy on GSM8K increases a lot. Both models have stable improvements on GSM8K and MATH. We will include the full results in the revised paper.

---

> ### Author Response · Authors · 2024-11-21
> **Response to Official Review by Reviewer kiSm (4/4)**
>
> **4. Number of Self-Edited Pairs**
>
> We appreciate this important observation and agree that further analysis is needed to explain the performance decline with more self-edited pairs. Our initial hypothesis is that introducing too many self-edited pairs may overwhelm the model with too many similar pairs, potentially leading to overfitting on certain patterns, since self-edited pairs from one problem share a large portion of context.
>
> To some extent, increasing self-editing pairs has a similar effect to increasing the number of training epochs under one self-editing pair, and more pairs will lead to overfitting. Thus, we may pre-set the number of self-edited pairs depending on the selection of training epochs and add more pairs if the model requires more training epochs under general DPO training.

---

> ### Author Response · Authors · 2024-11-24
> **Influence of Hyperparameter $\alpha$**
>
> We compare different values ​​of the hyperparameter $\alpha$. The results are shown as follows:
>
> | $\alpha$ | 0.01 | 0.05 | 0.1 |  0.2 |
> | :-----| :----: | :----: | :----: | :----: |
> | GSM8K | 88.5 | 88.4 | 87.9 | 87.7 |
> | MATH    | 59.3 | 59.9 | 59.6 | 59.3 |
>
> An excessively large $\alpha$ may reduce the model's generalization ability, which in turn results in lower accuracy on GSM8K and MATH.

---

> ### Author Response · Authors · 2024-11-25
> **Request for Feedback on Our Rebuttal**
>
> Dear Reviewer kiSm,
> \
> \
> We hope this email finds you well. Thank you again for your detailed review and valuable feedback on our submission.
>
> Since the rebuttal period is nearing its conclusion, we wanted to kindly follow up to request your feedback on our responses. We have worked diligently to address your concerns through additional experiments and detailed clarifications, and we would greatly appreciate hearing your thoughts on whether our rebuttal has resolved your questions.
>
> If there are any remaining issues or points requiring further clarification, we would be happy to address them before the rebuttal period ends. Thank you once again for your time and effort in reviewing our work.
> \
> \
> Sincerely,
>
> Authors of Paper #10957

---

> > ### Comment · Area_Chair_t9mg · 2024-11-26
> >
> > Dear reviewer kiSm,
> >
> > Could you please respond to authors' rebuttal and see if you would like to update your review? Thanks very much!
> >
> > AC

---

### Official Review · Reviewer_CRH2 · 2024-11-04

**Soundness:** 2
**Presentation:** 2
**Contribution:** 2
**Rating:** 5
**Confidence:** 3

**Summary:**

This paper proposes a preference learning framework called RISE for DPO training. The key idea is to inject carefully created noise in the correct-incorrect answer pairs to guide the model which mistakes to avoid.

**Strengths:**

The paper studies how to optimize preference learning, a timely and important topic. The proposed method RISE is easy to understand, and showcases empirical performance gains on two commonly used open-source models, qwen and llama.

**Weaknesses:**

My major concern is the scope of the performance evaluation. Across all experiments in this paper, the models are trained on a single dataset extracted from Lai et al 2024. It is not clear if the results are cherry-picked or not. For example, how about other DPO training datasets for Math and other datasets? In addition, it is not clear how RISE affects the models' quality on other tasks. The authors classify the 6 evaluation datasets used in this paper as either "in-distribution" or "out-of-distribution", but they are all indeed math-related questions. It would be more desired to see evaluation on real out-of-distribution datasets, such as physics or logic heavy datasets (e.g., ZebraLogic or ARC corpus).

**Questions:**

See my weakness comment.

---

> ### Author Response · Authors · 2024-11-21
> **Response to Official Review by Reviewer CRH2 (1/3)**
>
> Thanks a lot for your detailed review and constructive feedback. We appreciate your valuable comments and would like to address the concerns you raised:
>
> **1. Scope of Performance Evaluation and Dataset Selection**
>
> We understand the concern regarding the potential limitations of using a single dataset for training. Our choice of the Lai et al. (2024) dataset was motivated by two factors: (1) the good results achieved in that paper using this dataset, and (2) the convenience it provides in allowing direct comparison with methods in Lai et al. (2024).
>
> Nonetheless, we agree that evaluating on a broader set of datasets could provide more insight into the generalizability of our approach. To address this, we have implemented additional experiments using other mathematical datasets, including problems from the original training sets of the GSM8K [1] and MATH [2] datasets. We collect 15K problems like DART-math [3] to conduct RISE training. Preliminary results on Qwen2-7B-Instruct indicate that our RISE framework achieves better performance than the general DPO method:
>
> | Method                     | GSM8K | MATH |
> | :-----                         | :----:      | :----:    |
> | Qwen2-7B-Instruct  | 85.4       |  52.2    |
> | Qwen2-7B-DPO      | 87.7     |  57.5     |
> | Qwen2-7B-RISE      | **88.6**     |  **58.5**  |
>
> We will include more detailed results on other math evaluation datasets later.
>
> [1] Cobbe, Karl, et al. "Training verifiers to solve math word problems." arXiv preprint arXiv:2110.14168 (2021).
>
> [2] Hendrycks, Dan, et al. "Measuring mathematical problem solving with the math dataset." arXiv preprint arXiv:2103.03874 (2021).
>
> [3] Tong, Yuxuan, et al. "Dart-math: Difficulty-aware rejection tuning for mathematical problem-solving." arXiv preprint arXiv:2407.13690 (2024).

---

> ### Author Response · Authors · 2024-11-22
> **Response to Official Review by Reviewer CRH2 (2/3)**
>
> **2. Evaluation of other tasks**
>
> We appreciate the suggestion to include more diverse tasks in our evaluation. While the primary focus of our work was on math-related tasks, we agree that testing on truly out-of-distribution datasets would further strengthen the generalizability claims of our method.
> As suggested, we are expanding our evaluation to ZebraLogic (Puzzle Acc and Cell Acc), MBPP, and Humaneval to assess the model's performance on logic-based tasks and code generation. The results are shown as follows:
>
> |Method|Puzzle Acc|Cell Acc|MBPP|Humaneval|
> | :-----|  :----: | :----: |:----: |:----: |
> |Qwen2-7B-Instruct| 8.11|21.49|42.2|43.9|
> |Qwen2-7B-DPO| 8.10|20.82|42.0|45.1|
> |Qwen2-7B-RISE| **8.40**|**23.24**|**42.4**|**47.5**|
>
> We can observe that even without training on the above two tasks, our RISE outperforms both the original instruct model and DPO-tuned model. These results demonstrate the strong generalization of our RISE to out-of-domain tasks. The evaluation algorithm is based on two public evaluation repos [1] and [2].
>
> [1] https://github.com/bigcode-project/bigcode-evaluation-harness
>
> [2] https://github.com/WildEval/ZeroEval

---

> ### Author Response · Authors · 2024-11-24
> **Response to Official Review by Reviewer CRH2 (3/3)**
>
> **3. Adaptation to code generation**
>
> We apply our RISE method to code generation, where avoiding subtle errors is critical. Following [1], we adopt the LeetCode dataset [2] to conduct training. The dataset includes around 2K leetcode tasks in the medium and hard levels. For the Qwen2-7B-Instruct model, we sample 50 times and finally obtain 1473 pairs of chosen and rejected samples for training. The preliminary results are shown as follows:
>
> | Method | MBPP | Humaneval |
> | :-----| :----: | :----: |
> | Qwen2-7B-Instruct | 42.2 | 43.9 |
> | Qwen2-7B-DPO | 43.4 | 46.3 |
> | Qwen2-7B-RISE | **44.2** | **47.6** |
>
> We can observe that our RISE performs better than the general DPO method, achieving a 0.8% improvement on the MBPP test set and a 1.3% improvement on the Humaneval test set. Considering that the current solution editing strategy has not yet been adjusted to account for the characteristics of code generation, there is still certain room for improvement in the results.
>
> [1] Xu, Bin, Yiguan Lin, and Yinghao Li. "SRA-MCTS: Self-driven Reasoning Aurmentation with Monte Carlo Tree Search for Enhanced Code Generation." arXiv preprint arXiv:2411.11053 (2024).
>
> [2] https://huggingface.co/datasets/greengerong/leetcode

---

> ### Author Response · Authors · 2024-11-25
> **Request for Feedback on Our Rebuttal**
>
> Dear Reviewer CRH2,
> \
> \
> We hope this email finds you well. Thank you again for your thoughtful review and valuable feedback on our submission.
>
> As the rebuttal period is coming to an end, we wanted to kindly follow up to ask if you could provide any feedback on our rebuttal. We have worked hard to address your concerns and questions through detailed responses and additional experiments, and we would greatly appreciate your thoughts on whether our clarifications have sufficiently addressed your concerns.
>
> If there are any remaining issues needing further clarification, we would be more than happy to engage before the rebuttal period concludes. Thank you again for your time and consideration, and we truly appreciate your efforts in reviewing our work.
> \
> \
> Sincerely,
>
> Authors of Paper #10957

---

> > ### Comment · Area_Chair_t9mg · 2024-11-26
> >
> > Dear reviewer CRH2,
> >
> > Could you please respond to authors' rebuttal and see if you would like to update your review? Thanks very much!
> >
> > AC

---

> ### Author Response · Authors · 2024-12-02
> **Request for Feedback on Our Rebuttal**
>
> Dear Reviewer CRH2,
> \
> \
> I hope this message finds you well. As the rebuttal period is nearing its conclusion, we wanted to kindly follow up regarding your feedback on our submission. We highly value your insights and would greatly appreciate it if you could provide your response at your earliest convenience.
>
> We have worked diligently to address the concerns raised in the reviews and believe the updates and additional experiments we've conducted strengthen our submission significantly. We hope you might consider these improvements when evaluating the paper and its potential for a higher score.
>
> Thank you very much for your time and consideration. Please let us know if you have any further questions or concerns that we can address.
> \
> \
> Authors of Paper #10957

---

> ### Author Response · Authors · 2024-12-03
> **Request for Feedback on Our Rebuttal**
>
> Dear Reviewer CRH2,
> \
> \
> The rebuttal period ends in just a few hours, and we kindly ask for your feedback. We have carefully addressed your concerns and conducted additional experiments to strengthen our submission. We hope you will consider these improvements when reassessing the paper and its score.
>
> Thank you very much for your time and consideration!
> \
> \
> Authors of Paper #10957

---

### Author Response · Authors · 2024-11-25
**Official Comment by Authors**

Dear Reviewers,
\
\
We sincerely thank you for thoroughly assessing our work and providing us with valuable and constructive feedback.

Over the past few days, we have worked diligently to address your feedback and questions via additional experiments and detailed explanations. If you have any further queries or require clarifications, we would be happy to continue the discussion. We would also greatly appreciate it if you could reconsider the rating of our work based on the responses and updates we have provided during the rebuttal period.

Once again, we greatly appreciate your valuable contributions to improving our work.
\
\
Sincerely,

Authors of Paper #10957

---

### Author Response · Authors · 2024-11-26
**General Response**

We sincerely thank all the reviewers for their insightful and constructive feedback on our manuscript. We have carefully addressed each of the individual comments in the reviews and believe we have successfully responded to most of the concerns raised. Additionally, we have incorporated the suggested experiments, along with their discussions and results, in the revised manuscript.

Below, we provide a brief summary of the updates made in the revision, including: **(1) Core Contributions**, **(2) Strengths**, **(3) Updates During Rebuttal**

---

### **Core Contributions**

1. **Novel Framework**. We propose a novel preference learning framework that leverages the LLM itself to **inject errors** into correct solutions, constructing fine-grained hard pairs designed to **mitigate subtle errors**.
2. **Empirical Analysis**. Our study **identifies common error types** in mathematical reasoning and reveals the potential to improve the stability of the reasoning process by mitigating subtle errors.
3. **Experimental Results**. Through extensive experiments across various models (**Qwen2**, **Qwen2.5**, **Llama-3.1**, and **Ministral**) and tasks (**mathematical reasoning**, **logical reasoning**, and **code generation**), we demonstrate that RISE improves the reasoning capabilities of LLMs, helping LLMs mitigate subtle errors and consistently generate correct solutions.
4. **Transferability of Reasoning Ability across Diverse Domains.** Additional evaluation experiments on logical reasoning (ZebraLogic) and code generation (MBPP and Humaneval) show that our method can effectively generalize reasoning preferences learned from mathematical tasks to other complex reasoning domains even **without further in-domain training**.
5. **Flexible Adaptation to New Reasoning Tasks**. Our RISE framework has demonstrated its effectiveness in optimizing code generation, with **minimal modifications to editing prompts**. It shows that RISE can be flexibly and conveniently adapted to new tasks.

### **Strengths**

1. **Novelty of Method**. ``Reviewer kiSm`` and ``Reviewer r3F2`` agreed that our framework RISE is a novel and effective approach for preference learning.
2. **Significance**. ``Reviewer kiSm`` and ``Reviewer r3F2`` affirm that our study makes a contribution to the field of LLMs by focusing on subtle errors and improving mathematical reasoning ability.
3. **Writing and Presentation**. ``Reviewer CRH2`` and ``Reviewer nbYg`` praised the clarity and readability of our writing and presentation.
4. **Simplicity and Effectiveness**. ``Reviewer r3F2`` recognizes that our framework RISE is simple and effective.
5. **Self-improvement**. ``Reviewer r3F2`` approves that letting LLMs identify likely mistakes and optimizing based on error-injected self-edited pairs can lead to a self-improvement mechanism.

### **Updates During Rebuttal**

1. **Section 3.5**: Add evaluation results and analysis on out-of-domain reasoning tasks (logical reasoning and code generation).
2. **Appendix C**: Add validation experiments on more open-source models (Ministral-8B-Instruct and Qwen2.5-7B-Instruct).
3. **Appendix D**: Add validation experiments on another mathematical training dataset, including 15K problems from the original GSM8K and MATH training datasets.
4. **Appendix E**: Add exploration experiments of the hyperparameter $\alpha$.
5. **Appendix F**: Add exploration experiments of prompt designs, including self-instruct prompts and prompts that introduce arbitrary errors without specifying a particular mistake.
6. **Appendix G**: Add validation experiments on other reasoning tasks, such as code generation.

We believe these additions and clarifications comprehensively address the reviewers' concerns and enhance the overall quality of our manuscript.

---

> ### Author Response · Authors · 2024-12-04
> **Concerns Addressed During Rebuttal**
>
> In response to the reviewers' suggestions and concerns regarding our work, we have explained in detail and updated the manuscript in the following aspects:
> 1. **Scope of the performance evaluation** (``Reviewer CRH2``). We have conducted two additional experiments, (1) Training on another mathematical dataset (Appendix D) and (2) Out-of-domain evaluations without further training (Section 3.5). These experiments demonstrate that RISE **performs robustly** with different training datasets, and even achieves **strong transferability of reasoning ability across diverse domains**.
> 2. **Adaptation to other out-of-domain tasks** (``Reviewer kiSm``, ``Reviewer r3F2``). We have conducted experiments on code generation (Appendix G). The results indicate an efficient and effective adaptation to other complex reasoning tasks.
> 3. **Error Types** (``Reviewer kiSm``). We have conducted experiments with arbitrary error injection and different error combinations (Appendix F). The results illustrate the **scalability** of our framework.
> 4. **Validation on more models** (``Reviewer kiSm``). We have conducted experiments on Qwen2.5-7B-Instruct and Ministral-8B-Instruct (Appendix C). The results further demonstrate the **broader applicability** of our framework.
> 5. **Number of self-edited pairs** (``Reviewer kiSm``). Introducing too many self-edited pairs may lead to overfitting on certain patterns since self-edited pairs from one problem share a large portion of context.
> 6. **Hyperparameter $\alpha$** (``Reviewer kiSm``). We have explored the impact of the hyperparameter $\alpha$ in Appendix E.
> 7. **Performance on larger models** (``Reviewer nbYg``). We have conducted comprehensive experiments on Llama-3.1-70B-Instruct. The results suggest RISE’s robustness and effectiveness for improving reasoning in larger language models. Moreover, it performs better than the original DPO method.
> 8. **Prompt design** (``Reviewer r3F2``). We have conducted experiments with multiple prompt designs (Appendix F), (1) prompt template for introducing arbitrary errors and (2) self-instruct prompt template. The results indicate that our framework **performs robustly** and is not affected by variations in prompt design.

---

### Meta-Review · Area_Chair_t9mg · 2024-12-24

**Metareview:**

The paper presented an interesting method of generating hard negative preference pair construction through error-injected self-editing. The proposed method enhances the mathematical reasoning capability of LLMs by subtle error-aware DPO training.

Strength:
1. A useful method to generating hard negative examples to improve the reasoning capabilities of LLMs.
2. The method appeared to be novel, though the novelty might not be significant (An interesting negative sampling approach).

Weakness:
1. The improvement over existing methods is not very significant.
2. Method is heavily depending on prompt engineering and templates, which could limit its general applicability.

The paper is borderline. My major concern (similar to reviewer nbYg) is that the improvement on larger models is relatively minor (in the noisy region), signaling the method might not scale well. This could be seen in other tasks other than Math as questioned by reviewer CRH2.

**Additional Comments On Reviewer Discussion:**

The original support of paper is pretty lukewarm. Most of reviewers did not participate the discussion, unfortunately. In making the decision, AC considered the authors' rebuttal fully and found the overall improvement has not met the publication bar for ICLR.

---

### Decision · Program_Chairs · 2025-01-22

Reject